# Replica symmetry breaking in spin glasses in the replica-free Keldysh formalism

**Johannes Lang[1]\*, Subir Sachdev[2] and Sebastian Diehl[1]**

**1** Institut für Theoretische Physik, Universität zu Köln,
Zülpicher Straße 77, 50937 Cologne, Germany
**2** Department of Physics, Harvard University, Cambridge MA 02138, USA

\* [j.lang@uni-koeln.de](mailto:j.lang@uni-koeln.de)

## Abstract

We show that the algebra of Parisi ultrametric matrices is recovered by the real-time, replica-free, Dyson-Keldysh equations of infinite-range quantum spin glasses in the late time glassy limit. This connects to earlier results on classical and quantum systems showing how ultrametricity emerges from the persistent slow aging dynamics of the glass phase. The stationary spin glass state thereby spontaneously breaks thermal symmetry, or the Kubo-Martin-Schwinger relation of a state in global thermal equilibrium. We describe the Keldysh path integral of the infinite-range Ising model in transverse and longitudinal fields, and in the context of the Landau expansion of the action functional, show how the long-time limit connects to the full replica symmetry breaking obtained in the equilibrium formalism. We also illustrate our formalism by applying it to the spherical quantum $p$-spin model, which only exhibits one-step replica symmetry breaking.

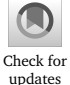
doi:10.21468/SciPostPhys.17.6.160

# 1  Introduction

The characteristic property of glasses is their slow evolution. As the system approaches the equilibrium state, its evolution becomes increasingly restricted by barriers in the free energy landscape [1–4] that take more and more time to overcome. As the time since the quench increases, relaxation slows down – the system 'ages'. One finds that accompanying this behavior is an ultrametric structure in the time dependence of correlations [5–10]. Because the dynamic constraints depend on the age of the glass, contrary to most other systems, it develops a sufficiently strong long-term memory for the age of the system to forever remain a relevant time scale [11]. Consequently, aging precludes glasses from reaching thermal equilibrium on accessible time scales [12–17].

Simultaneously, the analysis of the putative equilibrium state in systems with quenched disorder has brought forth many surprises, most prominently the breaking of replica symmetry [18–20], indicating the fragmentation of configuration space into disconnected energetically equivalent regions separated by insurmountable free energy barriers [2, 21, 22]. This fragmentation of the phase space breaks ergodicity [23] and gives rise to an ultrametric structure, observable in correlations [24].

Although theoretical research has focused largely on the simplest models exhibiting glassy behavior, namely spin systems with infinite-ranged interactions and quenched disorder, a connection to fragile glasses exists in mode coupling theory [25–28]. While lacking a rigorous derivation, numerical [29–32] and experimental evidence [33] support its conclusion that the characteristic properties of mean-field spin glasses carry over to systems with short-ranged interactions and annealed disorder in finite dimensions.

From the previous arguments, it is clear that aging dynamics and the absence of ergodicity are related phenomena [21]. In fact, in classical mean-field spin glasses, there have been numerous approaches to such connections [9, 10, 17, 34–40]. In this paper, we will describe the analogous connection in the Keldysh quantum formalism. Within this formalism we show that after a quench, at infinitely late times, the dynamic description eventually reproduces the equilibrium results, including the cases with ultrametricity and full replica symmetry breaking. It is important to point out that the infinite-time limit is taken at the beginning and the equilibrium result is not necessarily smoothly connected to any results obtained at finite times. In particular, the evolution never reaches the equilibrium state.

We summarize our main results in Sec. 2. There, we show that the algebraic properties of Parisi matrices, characterizing the fragmentation of configuration space, are recovered in the Keldysh formalism under the assumption of a strong hierarchy of time scales. The result then is applied in Sec. 3 to the quantum Sherrington-Kirkpatrick model in a longitudinal field and to the spherical quantum $p$-spin model in Sec. 4. Our approach exposes the spontaneous breaking of thermal symmetry (or the Kubo-Martin-Schwinger relation of a state in thermodynamic equilibrium) as the origin of replica symmetry breaking. This, however, is independent of the breaking of time-translation invariance as emphasized in Sec. 5. There we also apply constraints to the potential quantum critical scaling at zero temperature. In Sec. 6, we conclude

with an outlook discussing the connection to glasses of a finite age and to the zero-temperature limit.

# 2 Equivalence of ultrametric Keldysh dynamics and replica symmetry breaking

The proposal of a connection between replicas and the classical Langevin theory of spin glasses goes back to the classic early work of Sompolinsky and Zippelius [11, 34, 41]. They proposed that replica symmetry breaking was associated with multiple exponentially long time scales which diverged as the thermodynamic limit was taken. Later, Cugliandolo and Kurchan [5,6,42,43] showed that the classical equations exhibited 'aging' dynamics [44] in which the time scales remained finite, although exponentially long, even in the thermodynamic limit: they established an explicit connection between the aging equations and the replica symmetry breaking of the static problem. The connection to experimental dynamical observables was clarified in Refs. [7, 8], and further elaboration of the ultrametric case with full replica symmetry breaking was provided in Refs. [9, 10]. The aging dynamics was extended to quantum $p$-spin spherical models [40, 45, 46] and large $M_s$ SU($M_s$) quantum Heisenberg spin models [47] with one-step replica symmetry breaking using the Keldysh formalism: in the slow dynamics regime, the Keldysh equations became identical to the classical Langevin equations. The important case of the quantum Sherrington-Kirkpatrick Ising model, *i.e.* the Ising spin glass in a transverse field, was briefly discussed by Kennett *et al.* [48,49].

This section will present a general and model-independent discussion of the connection between replicas and glassy dynamics in the context of the Keldysh formalism. The analysis applies to quantum and classical spin glasses with possibly full replica symmetry breaking, including the recently studied quantum Ising spin glass in both transverse and longitudinal fields with an Almeida-Thouless transition [50], and the SU(2) quantum Heisenberg spin glass [51]. A related connection between supersymmetry and thermal symmetry in the paramagnetic phase of spin glasses was previously found by Kurchan [43] with applications to mode coupling theory discussed in Refs. [52–54].

The Parisi spin-glass order parameter, characterized by the function $p(u)$, $0 \leq u \leq 1$, in Eq. (1) is connected to the effective time-dependent (half) inverse temperature $X(t)$, defined by Eq. (12). The deviation of $X(t)$ from its equilibrium value $\beta/2$ measures the breaking of the fluctuation-dissipation relation by the glassy dynamics. For each $u < 1$, there is a $t$ which is determined by the solution of Eq. (17), where $\beta$ is the inverse temperature; smaller $u$ corresponds to larger $t$, with $u = 1$ mapping to $t = 0$ ($X(0) = \beta/2$), and $u = 0$ mapping to $t = \infty$ ($X(\infty) = 0$).

The analogy between the two approaches is complete in the sense that the algebra of ultrametric matrices in Eqs. (3), (4) and (5) is recovered by the real-time Dyson-Keldysh equations in the glassy limit under the assumption of ultrametricity in Eqs. (15), (20) and (22).

## 2.1 Replica formulation

On the replica side of the correspondence, we need to recall the algebraic relations satisfied by Parisi matrices, which we then aim to recover in the late-time limit of the dynamical equations.

For completeness, we begin by introducing the Parisi matrix $P_{ab}$ and the equivalent Parisi function $p(u)$ $u \in [0, 1]$. Note, that sometimes $u$ is called $x$ in the spin-glass literature. Now consider some model with $N$ replicas. Its equilibrium correlation functions are $N$-dimensional square matrices in replica space with an intriguing structure in the physical limit $N \to 0$. To capture this structure, we define that the $N$-dimensional symmetric matrix $P$ is called a Parisi

matrix if a sequence of integers $\mathcal{N} = \{n_1, n_2, \ldots, n_{L-1}\}$ with $n_1 = 1$ exists such that $P_{ab} = p_i$ for $\left[\frac{a-1}{m_i}\right] \neq \left[\frac{b-1}{m_i}\right]$, but $\left[\frac{a-1}{m_{i+1}}\right] = \left[\frac{b-1}{m_{i+1}}\right]$, where $m_i = \prod_{j=1}^{i} n_j$ and $m_L = N$. Furthermore, we fix the diagonal to $P_{aa} = p_0$. Simply put, a Parisi matrix consists of a hierarchy of block matrices placed along the diagonal such that each block itself is again a Parisi matrix (see Fig. 1(g)). If $P$ is identified with the correlation function, it describes the formation of clusters in replica space. If the $p_i$ form a decreasing sequence, different realizations of the system within a cluster are more strongly correlated with each other, than with replicas from other clusters. Thus, unless the sequence $\mathcal{N}$ contains only one element, the Parisi matrix describes the breaking of ergodicity.

The simple structure of $P$ allows it to be rewritten in terms of the equivalent Parisi function

$$p(u) = p_i, \quad \text{if} \quad m_i < u < m_{i+1}, \tag{1}$$

with $p(1) = p_0$. Since $m_1 = 1$ and $m_L = N$, with all other values in between, in the replica limit $N \to 0$ one has $u \in [0, 1]$, see Fig. 1(f). Note, that due to the limit $N \to 0$ smaller values of $u$ correspond to terms farther from the diagonal of $P$. Thus, inverting (1), $(dp(u)/du)^{-1}$ gives the probability of finding the value $p$ in the Parisi matrix. Again, if $P$ is interpreted as a correlation function, this determines the probability distribution of correlations between different realizations.

We define an ultrametric space as a metric space $M$ in which the triangle inequality is replaced by the strong triangle inequality

$$d_{ab} \leq \max\{d_{ac}, d_{cb}\}, \qquad \forall c, a, b \in M. \tag{2}$$

This implies that there are no points between $a$ and $b$, meaning that all points closer to $a$ than $b$ are at least a distance $d_{ab}$ from $b$. The space $M$ thus appears fractured into a hierarchy of clusters, such that on each level every point is a member of only one cluster [21]. If the $p_i$ are a decreasing sequence, replica space with the Parisi matrix $P$ as a measure of the inverse distance is ultrametric, meaning $P$ satisfies (2) with an appropriate choice for the dependence $d(P)$. One may choose for example $d_{ab} = 1/P_{ab}$ or $d_{ab} = p_0 - P_{ab}$. The hierarchical structure and the ultrametric condition can then both be read off in Fig. 1(g).

With these definitions, it immediately follows that the Hadamard (or component-wise) product of two Parisi matrices $A$ and $B$ is again a Parisi matrix $C$ with

$$c(u) = a(u)b(u). \tag{3}$$

Following some algebra (for a detailed derivation, see for example [55]), one finds that the same is true for matrix multiplication, for which one finds in the limit $N \to 0$

$$c(u) = a(u)b(1) + a(1)b(u) - u a(u)b(u) - \int_u^1 dv \, (a(u)b(v) + a(v)b(u)) - \int_0^u dv \, a(v)b(v). \tag{4}$$

Specifically, for the diagonal in replica space, the result simplifies to

$$c(1) = a(1)b(1) - \int_0^1 dv \, a(v)b(v). \tag{5}$$

Some intuition for the interpretation of the Parisi function can be gained by considering the unmagnetized ergodic case without replica symmetry breaking. In this case, the Parisi matrix $P$ is diagonal and therefore $p(u) \sim \delta_{1u}$, with $\delta_{ij}$ the Kronecker delta. In Fig. 1, this corresponds to the case with $p_1 = p_2 = p_3 = 0$. This is to be compared with a ferromagnetic or magnetized phase for which all $p_{n>0}$ are identical but non-zero. We point out that this ergodic solution

preserves replica symmetry as $P$ is invariant under permutations of its indices. Hence, although indistinguishable in terms of the Edwards-Anderson order parameter [56] $p_{EA} \equiv p(1^-) \equiv p_1$ alone, this gives a clear differentiation between the ferromagnetic and the spin-glass phase.

In general, Parisi functions are not continuous, and for practical purposes, it is often useful to write $p(u) = p_s(u) + p_f \delta_{1u}$, where $p_s(u)$ is continuous in the limit of $u \to 1$. In particular, $p_0 = p_1 + p_f$, $p_s(1) = p_1$, i.e., both $p_0$ and $p_s(1)$ involve the order parameter $p_1$. Due to the absence of aging in the ergodic phase, it is natural to expect at late times a relation between the off-diagonal terms $p_s(u)$ in replica space with the slow aging component of the evolution. Simultaneously, there should be a connection between the replica diagonal $p_f$ and the fast evolution that at late times becomes independent of the age of the system. In the following, we will show under which conditions these relations can be made rigorous.

## 2.2 Dynamic theory

We now show that the same rules of computing the Hadamard (or component-wise) product Eq. (3), and the matrix multiplication of two Parisi matrices Eq. (5) are also obtained from a dynamical Keldysh approach under the assumption of ultrametricity. The result is summarized in Tab. 1. We keep the details of the Keldysh formalism at a minimum; this allows us to discuss the connection to the Parisi algebra concisely. Later in the applications, we will start from the microscopic quantum models in the dynamical Keldysh formulation, and see how the objects discussed here arise in the course of calculation. This concerns the quantum Ising model with long-range disorder (Sec. 3), also known as the quantum Sherrington-Kirkpatrick model, and the quantum p-spin models 4.

A key object in the Keldysh formalism is the Green's function $G$, which can be organized in the following way (for an introduction, see [57]):

$$G = \begin{pmatrix} G^K & G^R \\ G^A & 0 \end{pmatrix}. \tag{6}$$

Here, the so-called Keldysh Green's function $G^K$ describes the correlations in the system (see Eq. (7) below for a spin system), and the retarded Green's function $G^R$ describes the retarded response to an external perturbation (see Eq. (11) below). The advanced Green's function is related to the retarded by $G^A = (G^R)^\dagger$ and appears naturally in the calculation of virtual processes.

In the infinite-range mean-field models considered throughout this work, the two-point Green's function has no spatial dependence. The relevant low energy degrees of freedom are coarse-grained, collective real scalar spin variables $S$. The Green's function then merely depends on two times, $G = G(t_1, t_2)$. For example, the Keldysh Green's function for the collective spin variable is

$$\langle S(t_1)S(t_2)\rangle = G^K(t_1, t_2). \tag{7}$$

An alternative parameterization of time variables is in Wigner coordinates (see e.g. [57]), introducing center-of-mass and relative time,

$$T = (t_1 + t_2)/2, \quad \text{and} \quad t = t_1 - t_2, \tag{8}$$

see Fig. 1(a). We will consider the dynamics after a quench at time $T = 0$. The strict connection to the Parisi algebra follows when we send the time passed since the quench $T \to \infty$. In this limit, the center-of-mass time becomes but an overall scale that drops out; this ingredient will be used here as an assumption, and be justified from the explicit microscopic model calculations in Secs. 3, 4. We thus transform to Wigner coordinates and study the limit

$$G(t_1, t_2) = G(T, t) \underset{T \to \infty}{\longrightarrow} G^K(t). \tag{9}$$

To complete the physical setup studied throughout this work: We consider isolated systems with an energy density corresponding to an inverse temperature $\beta$ in equilibrium, including the zero temperature quantum limit $\beta \to \infty$. This does, of course, not immediately imply that the system globally equilibrates to this temperature; the actual state of the system has to be found in a problem-specific way from solving the Dyson equation, i.e., the equations of motion for the Green's function (see Secs. 3, 4). For example, for the random quantum Ising model, this equation reduces to a standard Boltzmann equation in the limit of vanishing randomness, describing global thermalization. Including randomness leads to corrections that compete with thermalization, and give rise to the stationarity condition of the quantum Sherrington-Kirkpatrick model, which does not always thermalize but can show glassy behavior.

We then split the the Green's functions into a fast part that equilibrates at late times, and a slow part that describes aging

$$G(t) = G_s(t) + G_f(t) \equiv \int_{-\Lambda/b}^{\Lambda/b} d\omega\, e^{-i\omega t} G(\omega) + \underbrace{\int_{\Lambda/b}^{\Lambda} d\omega\, e^{-i\omega t} G(\omega) + \int_{-\Lambda}^{-\Lambda/b} d\omega\, e^{-i\omega t} G(\omega)}_{=G_f(t)}. \tag{10}$$

Here $\Lambda$ is a high-frequency cutoff. From the view of the scales of the aging variables (index $s$) $b \ll 1$, whereas $b$ can be sent to 1 from the view of the fast variables (this makes sense particularly when the scale separation between aging and stationary field diverges with $T \to \infty$). By this construction, the fast field has support in the time domain on scales $b/\Lambda \leq |t| \leq 1/\Lambda$, and the slow one varies with time for $|t| \geq b/\Lambda$, while it is constant for $|t| \leq b/\Lambda$. We therefore identify $b/\Lambda = \tau_{\mathrm{erg}}$ as the time scale up to which correlations are ergodic. Furthermore, we anticipate $G_f$ and $G_s$ for $t \sim b/\Lambda$ to correspond to $p_f$ and $p_1$ for the appropriate Parisi function $p(u)$. The emergence of the scale $\tau_{\mathrm{erg}}$ for finite $T$ implying imperfect separation between $G_f$ and $G_s$ is shown in Fig. 1(b). The boundary value $G^K(t = 2T)$ vanishes as $T \to \infty$.

Next, we address the response to an external (longitudinal) field $h$ which is given by the retarded Green's function

$$G^R(t_1, t_2) = \frac{\delta \langle S(t_2) \rangle}{\delta h(t_1)}. \tag{11}$$

Since the advanced Green's function for real scalar theories can be expressed as $G^A(t_1, t_2) = G^R(t_2, t_1)$, the dynamic theory can be formulated in terms of the two real functions $G^R(t_1, t_2)$ and $G^K(t_1, t_2)$. In the dynamical formalism of classical spin glasses, these are commonly referred to as $G$ and $C$ respectively. Due to causality, the former vanishes for negative relative times $t < 0$. In the limit $T \to \infty$, it can therefore be written in the form of a generalized thermal ansatz [6]

$$G_s^R(t) = -X(t)\theta(t)\partial_t G_s^K(t) = G_s^A(-t), \tag{12}$$

where $\theta(t)$ is the Heaviside function and $X(t)$ plays the role of a time-dependent inverse temperature in the high-temperature expansion of the fluctuation-dissipation relation

$$G^R(t) = -\theta(t)\tan\left(\frac{\beta}{2}\partial_t\right) G^K(t)$$
$$= -\theta(t)\frac{\beta}{2}\partial_t G^K(t) + \mathcal{O}(\beta^2). \tag{13}$$

We emphasize that this interpretation becomes particularly meaningful at late times when the characteristic time scales of the evolution satisfy $\Delta t \gg \beta$, which justifies the expansion in powers of the inverse temperature $\beta$.

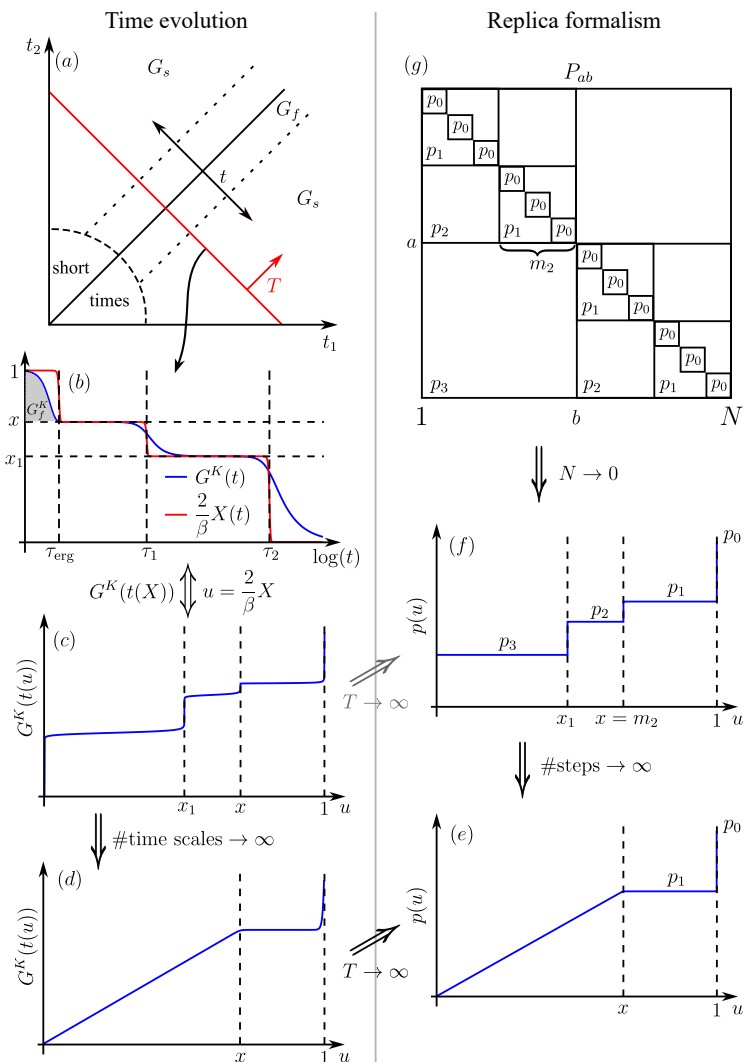

Figure 1: Illustration of the structural similarity between the dynamical theory at asymptotically late times and replica theory. (a) For the time domain, we work in Wigner coordinates $T, t$. (b) Correlation function $G^K$ and dynamical temperature $X$ along a cut with fixed $T$. We parametrize the fields into fast $G_f$ (gray shaded area) and slow fields $G_s$, for short and long relative times $t$. (c) The monotonic function $X(t)$, defined in Eq. (12), is used to map time to a compact domain. Since $X(t)$ is decreasing, small values of $u$ correspond to large $t$. (d) As the number of time scales $\tau_n$ are sent to infinity, $G^K(u)$ becomes a smooth function. (e),(f) In the limit $T \to \infty$, $G^K(u)$ is structurally identical to a Parisi function obtained in the limit $N \to 0$ from a Parisi matrix. (g) Parisi matrix for $N = 12$ in the specific case with $m_2 = 3$, $m_3 = 6$ and $m_4 = N$. We show the case of 2-step replica symmetry breaking, except for (d) and (e), which demonstrate the extension to full replica symmetry breaking corresponding to an infinite number of time scales or equivalently layers in the Parisi matrix. In this case, the asymptotic correlation functions are ultrametric, such that the equilibrium result is indeed eventually reached by the dynamics. This is in general not the case for $n$-step replica symmetry breaking with $n$ finite. Although $G^K(u)$ in (c) reproduces the Parisi function $p(u)$ in (f), the absence of ultrametricity of $G^K(t)$ means that they do not satisfy the same algebra. This is indicated by the gray arrow between (c) and (f).

Table 1: Translation table between replica formalism and ultrametric Keldysh theory. Here, $A$ and $B$ are Parisi matrices evaluated in the limit $N \to 0$. The corresponding Parisi functions are $a(u)$ and $b(u)$ with $u \in [0,1]$. $A^{K/R}(t)$ etc., are the associated ultrametric correlation/response functions in relative time $t$ and $\circ$ denotes their convolution.

| replica theory | ultrametric Keldysh theory |
|---|---|
| $A_{ab}$ | $A^K(t)$ |
| $A_{aa} = a(1) = a_s(1) + a_f$ | $A^K(t=0) = A^K_s(0) + A^K_f(0)$ |
| $A_{ab}B_{ab}$ | $A^K(t)B^K(t)$ |
| $(A \cdot B)_{ab}$ | $(A^K \circ B^A + A^R \circ B^K)(t)$ |

From its definition (7), it is clear that $G^K(t_1, t_2)$ is symmetric under exchange of $t_1$ and $t_2$. Without loss of generality, the dynamic theory can therefore be restricted to $t > 0$. At short relative times ($t < \tau_{\mathrm{erg}}$), all correlations have equilibrated, which fixes the boundary condition $X(0) = \beta/2$. But at large values of $t$ that diverge as the inverse infrared cutoff $T$ is sent to infinity, the system becomes increasingly fragmented and thus unresponsive. Hence, we expect the correlations to decay slowly in $t$ and $X$ to be a decreasing function of $t$, see Fig. 1(b). The ansatz (12) is consistent with that of Ref. [6] and a generalization of the one used in Ref. [39]. From the expansion (13) it also follows that the ansatz (12) corresponds to a restriction of the Parisi function to the first Matsubara frequency in the equilibrium approach. Due to the exceedingly slow dynamics in the aging regime, following the same argument as in the expansion in Eq. (12) this ansatz becomes exact in the limit $T \to \infty$ at any finite temperature.

Finally, we make the assumption of strong hierarchy [34], which is to say that correlations vary so slowly in time that $G^K_s(t) < G^K_s(t')$ requires $\lim_{T \to \infty} t/t' = 0$. This implies that correlations are ultrametric since

$$G^K_s(t + t') = G^K_s(\max(t, t')), \tag{14}$$

satisfies the strong triangle inequality $G^K_s(t) \geq \min\{G^K_s(t - \tau), G^K_s(\tau)\} \; \forall \tau \in \mathbb{R}$. Each value of $G^K_s$ can be assigned to a characteristic time scale. In the case of infinitely many time scales Eq. (14) is also the only dependence on relative time $t$ in the limit of $T \to \infty$ that is consistent with aging dynamics [6]. Conversely, if only a finite number of time scales emerges, this is not expected to hold true for the late-time dynamics [5]. We will see below in Sec. 4, that this implies that in the thermodynamic limit a quench in the spin-glass phase of the spherical $p$-spin model never reaches thermal equilibrium.

With these preparations, we can now consider the product of two Green's functions in the time domain. We focus on the Keldysh component

$$C^K(t) = A^K(t)B^K(t) = A^K_s(t)B^K_s(t) + A^K_f(t)B^K_f(t). \tag{15}$$

In the examples below, we will show how these products enter the equation of motion for the Keldysh Green's function as a result of memory effects. We can restrict to the product of Keldysh components: products involving retarded/advanced components $A^{R/A}, B^{R/A}$ can be reduced to those using Eq. (12). We identify the equal-time expression $C^K(t=0)$ with the replica diagonal, i.e. equilibrated, part of the Parisi function $c(1)$, and the slow component $C^K_s(t)$ with the off-diagonal parts describing replica symmetry breaking $c(u)$ with $u \in [0, 1[$. The product of two Keldysh Green's functions in the time domain is therefore equivalent to the Hadamard product of two Parisi matrices (see Table 1, which summarizes our key results).

We next show that this correspondence also extends to convolutions. Here, the relevant combination of Green's functions, which appears ubiquitously in the contributions to the Dyson equation of motion due to disorder averaging and describes memory effects, is of the form $A^K \circ B^A + A^R \circ B^K$, with $A \circ B = \int_{t'} A(t - t')B(t')$. We then split again into slow and fast components, $A = A_s + A_f, B = B_s + B_f$.

We first consider the product of the slow components

$$
\begin{aligned}
A_s^K \circ B_s^A + A_s^R \circ B_s^K = & -\int_0^t dt' A_s^K(t + t')X(t')\partial_{t'}B_s^K(t') - \int_t^\infty dt' A_s^K(t + t')X(t')\partial_{t'}B_s^K(t') \\
& -\int_0^t dt' B_s^K(t - t')X(t')\partial_{t'}A_s^K(t') - \int_t^\infty dt' B_s^K(t - t')X(t')\partial_{t'}A_s^K(t') \\
= & \int_{X(0)}^{X(t)} dX' A_s^K(X(t))B_s^K(X') - A_s^K(X(t))X(t)B_s^K(X(t)) \\
& + \int_{X(0)}^{X(t)} dX' B_s^K(X(t))A_s^K(X') - B_s^K(X(t))X(t)A_s^K(X(t)) \\
& + A_s^K(X(t))X(0)B_s^K(X(0)) + B_s^K(X(t))X(0)A_s^K(X(0)) \\
& + A_s^K(X(t))X(t)B_s^K(X(t)) + \int_{X(t)}^0 dX' A_s^K(X')B_s^K(X') \\
= & \frac{\beta}{2}\left[ -\int_u^1 dv\, A_s^K(u)B_s^K(v) - u A_s^K(u)B_s^K(u) + A_s^K(u)B_s^K(1) \right. \\
& \left. -\int_u^1 dv\, B_s^K(u)A_s^K(v) + B_s^K(u)A_s^K(1) - \int_0^u dv\, A_s^K(v)B_s^K(v) \right]. \quad (16)
\end{aligned}
$$

In the first equality, we have used the generalized thermal ansatz. It implies that the glass phase becomes stiff as $T \to \infty$: Since $G_s^K(t)$ decays on a time scale $t \sim T$, the derivative scales as $\partial_t \sim 1/T$ and compensates the divergence of the integration domain $\sim T$. This behavior is illustrated in Fig. 2. We point out that this stiffness property of the classical ansatz (12) ensures a weak long-term memory and thus convergence of the convolutions even in the aging regime. A more responsive, i.e. more slowly decaying $G^R$, would imply a stronger memory and divergent convolutions in Eq. (16) while for a less responsive ansatz, the integrals vanish thereby precluding glassy behavior. The second equality in Eq. (16), which compactifies time, follows from ultrametricity and partial integration. It is important to point out that due to this change of variables, the information on time scales is lost. In the last step, we have introduced the dimensionless variable $u \in [0, 1]$ as

$$
X(t) = \beta u/2, \quad (17)
$$

with the boundary conditions $X(0) = \beta/2$ and $X(\infty) = 0$. The same relation holds between $v$ and $X'$. $X(t)$ is a decreasing function. Consequently, small values of $u$ correspond to late times $t$, and while the system equilibrates at short relative times, $X(\infty) = 0$ implies a maximally unresponsive infinite temperature state at large relative times.

As a consequence of the stiffness implied by the generalized fluctuation-dissipation relation (12) with $X(t) \le \beta/2$ the slow retarded Green's function decays faster than the slow Keldysh component and can therefore be neglected at sufficiently late times $t$ (see also Fig. 2). Consequently, one finds

$$
A_f^K \circ B_s^A + A_s^R \circ B_f^K = 0, \quad (18)
$$

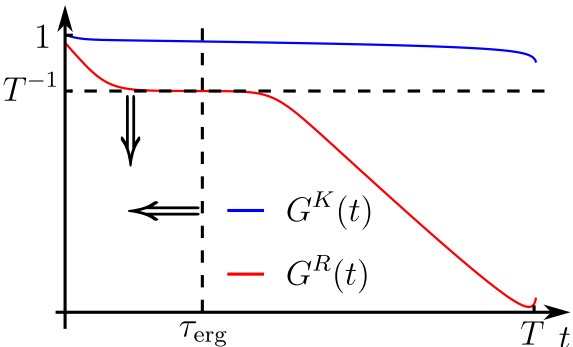

Figure 2: Stiffness of the glass phase. We show a logarithmic plot of typical correlation and response functions $G^K(t)$ and $G^R(t)$ at intermediate center-of-mass times $T$. In the aging regime $t > \tau_{\text{erg}}$ the correlation function $G^K(t)$ varies slowly, such that the generalized thermal ansatz Eq. (12) implies that $G^R(t \gtrsim \tau_{\text{erg}}) \lesssim 1/T$ vanishes as $T \to \infty$. The arrows indicate this behavior of the response function as $T$ is increased. For times $t < \tau_{\text{erg}}$ the system is in thermal equilibrium. The boundary effects for $t \approx T$ that cause $G^R$ to rise quickly become irrelevant as $T \to \infty$.

while the other term mixing fast and slow parts is finite

$$
\begin{aligned}
A_s^K \circ B_f^A + A_f^R \circ B_s^K &= \int_t \left[ B_f^R(t) A_s^K(u) + A_f^R(t) B_s^K(u) \right] \\
&= \frac{\beta}{2} \left[ B_f^K(1) A_s^K(u) + A_f^K(1) B_s^K(u) \right].
\end{aligned}
\tag{19}
$$

The first line follows from the condition of strong hierarchy: The slow parts are constant on the scale on which the fast functions decay. To obtain the simplified expression in the second line, we have used the high-temperature expansion of the standard fluctuation-dissipation relation Eq. (13) to linear order in $\beta$ for the fast field, which makes the analogy to the replica formalism more apparent. It will therefore be used throughout this article. We emphasize, however, that it is not essential to the argument. Combining all terms, we find

$$
\begin{aligned}
A^K \circ B^A + A^R \circ B^K = \frac{\beta}{2} \Bigg[ & A_s^K(u) B^K(1) + B_s^K(u) A^K(1) - u A_s^K(u) B_s^K(u) \\
& - \int_u^1 dv \left( A_s^K(u) B_s^K(v) + B_s^K(u) A_s^K(v) \right) - \int_0^u dv A_s^K(v) B_s^K(v) \Bigg],
\end{aligned}
\tag{20}
$$

which is to be compared with Eq. (4). In Eq. (20) that fast fields $B_f$ and $A_f$ enter only via $B^K(u=1)$ and $A^K(u=1)$ in the first two terms as is the case for the replica diagonal in Eq. (4).

We are left with the task of evaluating the time diagonal in the Keldysh formulation. Sending $t \to 0$ and using the same arguments as above, we find

$$
\begin{aligned}
A_f^K \circ B_s^A + B_f^K \circ A_s^R \Big|_{t=0} &= 0, \\
A_f^K \circ B_f^A + B_f^K \circ A_f^R \Big|_{t=0} &= \frac{\beta}{2} A_f^K(1) B_f^K(1), \\
A_s^K \circ B_s^A + B_s^K \circ A_s^R \Big|_{t=0} &= \frac{\beta}{2} \left[ A_s^K(1) B_s^K(1) - \int_0^1 dv A_s^K(v) B_s^K(v) \right], \\
A_s^K \circ B_f^A + B_s^K \circ A_f^R \Big|_{t=0} &= \frac{\beta}{2} \left[ B_f^K(1) A_s^K(1) + A_f^K(1) B_s^K(1) \right].
\end{aligned}
\tag{21}
$$

Putting everything together, this gives for the time-diagonal

$$A^K \circ B^A + A^R \circ B^K \Big|_{t=0} = \frac{\beta}{2} \left[ A^K(1)B^K(1) - \int_0^1 dv A_s^K(v) B_s^K(v) \right]. \tag{22}$$

Comparison with Eq. (5) shows that the matrix multiplication in Keldysh formalism at asymptotically late times using ultrametricity and a generalized thermal ansatz in the classical limit is identical to the matrix multiplication in the replica formalism.

As has previously been reported by Cugliandolo and Kurchan [5], this approach also gives an interpretation of the replica average in equilibrium theory. For example, the averaged correlation function

$$Q_\infty = \int_0^1 du \, q(u), \tag{23}$$

with $q(u)$ the Parisi function of the replica matrix $Q_{ab} = \langle s_i^a s_i^b \rangle$, is related to the integrated response function

$$Q_\infty = 1 + \int_0^\infty dt \, X(t) \partial_t Q^K(t) = 1 - \int_0^\infty dt \, Q^R(t). \tag{24}$$

In summary, we have shown that, under the assumption of ultrametricity, the Keldysh component of convolution integrals in time $A \circ B$ reproduces the algebra of replica matrices in the limit $N \to 0$. Since the replica Fourier transform satisfies the convolution theorem [58] a similar equivalence can be found for products in frequency space and replica Fourier space.

# 3 Application: The quantum Sherrington-Kirkpatrick model

We now turn our attention to the most general case of replica symmetry breaking, known as full RSB and realized by the Sherrington-Kirkpatrick model. We begin with the derivation of the Landau action valid near the critical point. The procedure can be understood as the out-of-equilibrium version of the Landau theory presented in Ref. [50]. Our approach is similar in spirit to that of Sompolinsky and Zippelius [11, 34, 41] that culminated in the analytical solution of the late-time relaxation obtained by Cugliandolo and Kurchan [6]. Following several attempts at recovering the results of replica theory from the dynamical equations at late times [34–37] long-standing discrepancies were resolved in Ref. [10]. There remains, however, a conceptual difference between the two approaches arising from the order of limits [43]. If one considers a finite system at infinitely late times and eventually sends the system size to infinity [34], the system is time-translation invariant, but must also obey the fluctuation-dissipation relation [59], as at infinite times, any finite system is fully equilibrated [17]. Consequently, a violation of the fluctuation-dissipation relation in this limit contradicts the underlying assumptions. The thermal symmetry cannot be broken spontaneously. Considering instead an infinite system at late but finite times [6], time-translation invariance is always broken because the equilibration time is determined by the system size and therefore never reached. Our approach considers an infinite system from the outset and then sends time to infinity, which allows us to study the spontaneous breaking of thermal symmetry. By measuring time in terms of the inverse temperature of the generalized fluctuation-dissipation relation we have access to all relevant infinite time scales.

Recent experimental developments have resulted in renewed interest in spin glasses. In particular, the precise positioning of large numbers of Rydberg atoms with tweezers provides an avenue towards the realization of spin glasses with long-ranged interactions [60–65]. The idea is as follows, lasers are used to drive the Rabi transition between ground-state atoms and a highly excited long-lived Rydberg state. As no other states get occupied, it is sufficient to describe the atoms as two-level systems that interact via van-der Waals interactions only when in the large and therefore highly polarizable Rydberg state. By positioning the atoms at random but fixed sites using optical tweezers, the strengths of the interactions are randomized [61]. Finally, the occupation of the Rydberg states can be controlled by adjusting the detuning $\delta$ of the driving laser, which leads to a longitudinal field $h = \delta$ in the effective spin model [66]

$$H = \sum_{ij} J_{ij} Z_i Z_j - \sum_i X_i - h \sum_i Z_i \,, \tag{25}$$

where $X_i, Z_i$ are the Pauli operators on qubits at site $i$. Here the Rabi coupling has been set to one and $J_{ij}$ denotes the van-der Waals interaction between atoms $i$ and $j$.

We point out that other experimental schemes such as Rydberg dressing which uses lasers far detuned from the Rabi transition to increase the lifetime at the expense of weaker interactions or microwave coupling between different Rydberg states leading to longer-ranged interactions $\sim R^{-3}$ result in the same Hamiltonian [66, 67]. Furthermore, random long-range interactions can be achieved with quantum simulators based on superconducting qubits [68] or by trapping atoms in a confocal cavity [69, 70]. Although in the latter case, the driven-dissipative cavity prevents the system from reaching thermal equilibrium, significant similarities with the classical Sherrington-Kirkpatrick model have been found in theory [71, 72] and experiment [73].

An important distinction between the new platforms and classical glasses is the finite lifetime of the excited states due to spontaneous emission. It is therefore important to develop a minimal dynamical description applicable to late but finite times. In the following, we thus first derive the Ginzburg-Landau effective action for the quantum Sherrington-Kirkpatrick model (25) near the critical point where the spin glass forms. The Dyson-Keldysh equations obtained from it are then analyzed in the ultrametric limit and shown to reproduce the full replica-symmetry-breaking solution of the equilibrium model.

## 3.1 Effective action

To obtain the equations of motion, we derive the effective Ginzburg-Landau action of the random Ising model in a longitudinal and a transversal field as defined in (25). Without loss of generality, the quenched disordered coupling strengths $J_{ij}$ are drawn from a Gaussian distribution independent of the site indices $i$ and $j$. Hence, this model first introduced in Ref. [74], is effectively infinite-dimensional and described by mean-field theory. Its equilibrium Landau action has been studied in Ref. [75], with aging dynamics analyzed in Ref. [49] and previously based on the approach of Sompolinsky and Zippelius [11, 41] in Ref. [6]. Near the phase transition, we can average the spins over a small domain, and integrate over the transverse spin components, such that the discrete spins are replaced by a real bosonic variable $S_i \sim Z_i$; in this process the spin Berry phase, which has a first-order time derivative, is replaced by a kinetic term which has a second-order time derivative [75] (in other words, Ising spins in a transverse field are similar to quantum rotors). Integrating out the disordered coupling

strengths $J_{ij}$, the site index can be dropped, and the effective action is given by [49]

$$s[S] = s_0[S] + s_g[S] + s_\kappa[S] + s_h[S],$$

$$s_0[S] = -\frac{1}{2}\int_t \sum_\eta \eta S_\eta(t)[\partial_t^2 + m^2]S_\eta(t),$$

$$s_h[S] = \int_t \sum_\eta \eta h_\eta(t)S_\eta(t),$$

$$s_g[S] = -\frac{g}{2}\int_t \sum_\eta \eta S_\eta^4(t),$$

$$s_\kappa[S] = i\frac{\kappa}{4}\int_{t_1,t_2} S_\eta(t_1)\sigma_{\eta\rho}^3 S_\rho(t_1)\, S_{\eta'}(t_2)\sigma_{\eta'\rho'}^3 S_{\rho'}(t_2).$$

(26)

With the spins represented by the bosonic variable $S$, Pauli matrices here and in the following act exclusively on the Keldysh/time-contour space. The effective mass $m$ tunes between the paramagnetic and spin-glass phase. As noted above, the transverse field gives rise to the inertial dynamic term in $s_0$. The quartic term $s_g$ provides a soft constraint for the spin length. Replacing the hard- by a soft-spin constraint is possible since we only target the description of low frequency modes as appropriate for glasses within our Landau effective field theory approach. Its key assumption is that, since the expectation value of $Q$ vanishes on the paramagnetic side, we can expand in small fluctuations, taking only low order powers into account, which are compatible with the symmetries of the problem. The same terms will be encountered irrespective to whether we start, on the microscopic level, from a hard-spin model or from a softened constraint. While either choice will lead to different non-universal coupling parameters for the Landau theory, both give rise to the same universal low frequency behavior [75]. The non-linearity gives rise to an interacting impurity model, which has to be treated perturbatively. A stable Landau theory obtains when expanding to second order in $g$, see Sec. 3.3.

The disorder is encoded in the term $s_\kappa$, with $\kappa = \bar{J}_{ij}^2$ the variance of the Gaussian distribution $\mathcal{P}(J_{ij})$. The dynamical theory requires a doubling of the time contour. Following the standard procedure of the Keldysh path-integral (for an introduction see [57,76]), we therefore introduce Greek indices that take the values $\{+,-\}$ to denote the branch of the contour. Although external fields are classical, we keep the notation symmetric and thus distinguish between the longitudinal field $h$ on the forward $(+)$ and backward $(-)$ branch.

Due to the infinite range of the random couplings $J_{ij}$, the site index in (26) is irrelevant and will be suppressed in the following.

It is convenient to introduce the spin bilinear $q_{\alpha\beta} \equiv q_{\rho\rho'}(t_1, t_2) = S_\eta(t_1)\sigma_{\eta\rho}^3 S_{\rho'}(t_2)$. Here and in the following $\alpha$ and $\beta$ denote multi-indices incorporating the Keldysh index and time. We now decouple the disorder-induced non-linearity by a Hubbard-Stratonovich transformation with

$$s_{\text{aux}}[S,Q] = \frac{i}{4\kappa}\text{Tr}\left[(RQ\sigma^1 R - i\kappa q)^2\right],$$

(27)

where $R = (\sigma^1 + \sigma^3)/\sqrt{2}$. We have furthermore introduced the trace Tr over the multi index, i.e. $\text{Tr}[A^2] = \int_{t,t'} A_{\eta\rho}(t,t')A_{\rho\eta}(t',t)$. Rewriting the soft constraint $s_g$ in terms of a functional derivative with respect to a source field $K$, we can perform the remaining Gaussian integral over $S_\eta$. Rotating the result to the $R/A/K$ basis, we define classical and quantum fields as $h_c = \sum_\eta h_\eta/\sqrt{2}$ and $h_q = \sum_\eta \eta h_\eta/\sqrt{2}$ to express the Keldysh partition function as

$$Z = \int \mathcal{D}Q\, e^{is_g[\frac{\delta}{\delta K}]} e^{-\frac{i}{2}(K^\top + h^\top)\sigma^1[G_0^{-1}+Q]^{-1}\sigma^1(K+h)} e^{-\frac{1}{4\kappa}\text{Tr}[Q\sigma^1 Q\sigma^1] - \frac{1}{2}\text{Tr}\log(\mathbb{1}+G_0 Q) + \text{const.}}\bigg|_{K=0}.$$

(28)

Here, $\top$ denotes the transpose in Keldysh space. Finally, the bare spin propagator can be expanded in powers of $m^{-2}$

$$
\begin{aligned}
G_0(t_1, t_2) &= -\delta(t_1 - t_2)\sigma^1 \left(\partial_{t_2}^2 + m^2\right)^{-1} \\
&\approx \delta(t_1 - t_2)\sigma^1 \left(-\frac{1}{m^2} + \frac{1}{m^4}\partial_{t_2}^2\right).
\end{aligned}
\tag{29}
$$

Due to the saddle point condition

$$
0 = -2i\kappa \frac{\delta s_{\text{aux}}}{\delta Q_{\alpha\beta}} = (\sigma^1 Q \sigma^1 - i\kappa \sigma^1 RqR)_{\beta\alpha},
\tag{30}
$$

the average of $Q$ can be given a physical interpretation in terms of the full spin propagator $G = \left(G_0^{-1} + Q\right)^{-1}$

$$
\frac{1}{\kappa}\langle Q_{\alpha\beta}\rangle = i\langle RqR\sigma^1\rangle_{\alpha\beta} = -(\sigma^1 G \sigma^1)_{\alpha\beta}.
\tag{31}
$$

It is however not a valid order parameter as it does not vanish at the critical point. We arrange for this property by a shift operation, which can also be viewed as a UV renormalization operating on short time distances. Thereby, we fix the value of the critical coupling strength via the requirement that the renormalized order parameter vanishes at the transition, which will be verified explicitly below. To this end, we decompose $Q(t, t') = \left((m^2 - \tilde{m}^2)\sigma^1 + Q_{\text{EA}}\right)\delta(t - t') + \tilde{Q}(t, t') \equiv Q_0(t, t') + \tilde{Q}(t, t')$ into a UV shift $Q_0(t, t')$ with $Q_{\text{EA}} = iq_{\text{EA}}(\mathbb{1} - \sigma^3)/2$ and a small field $\tilde{Q}$. We also define $\tilde{G}_0^{-1} = G_0^{-1} + Q_0$, use again the exact relation Eq. (31), and expand in powers of $\tilde{Q}$

$$
\begin{aligned}
\frac{1}{\kappa}\left[\sigma^1 \left(Q_0 + \tilde{Q}\right)\sigma^1\right]_{\alpha\beta} - i\left[\tilde{G}_0 \sigma^1 (hh^\top)\sigma^1 \tilde{G}_0\right]_{\alpha\beta} &= -\left[(\tilde{G}_0^{-1} + \tilde{Q})^{-1}\right]_{\alpha\beta} \\
&\approx \left[-\tilde{G}_0 + \tilde{G}_0 \tilde{Q}\tilde{G}_0 - \tilde{G}_0 \tilde{Q}\tilde{G}_0 \tilde{Q}\tilde{G}_0 + \dots\right]_{\alpha\beta}.
\end{aligned}
\tag{32}
$$

We approximate $\tilde{G}_0 \approx -\tilde{m}^{-2}\sigma^1 \delta(t - t')$ everywhere except for the zero-order term in $\tilde{Q}$, where we also expand in the time derivative to leading order. Furthermore, we make use of the fact that the magnetic field is classical and time-independent. With this, the term due to the magnetic field simplifies to $i[\tilde{G}_0 \sigma^1 (hh^\top)\sigma^1 \tilde{G}_0]_{\alpha\beta} = ih^2 \tilde{G}_0^R \tilde{G}_0^A \delta_{\alpha 1}\delta_{\beta 1} \approx \frac{ih^2}{\tilde{m}^4}\delta_{\alpha 1}\delta_{\beta 1}$. Most importantly, we notice, that both the magnetic field and the order parameter $q_{\text{EA}}$ only affect the Keldysh component of the matrix equation (32). This is an exact statement, that follows from the causal structure of the Green's function. It implies that the magnetic field can be absorbed into $q_{\text{EA}}$.

Explicitly, upon Fourier transformation, the Keldysh and retarded components of

$$
\tilde{Q} = \begin{pmatrix} \tilde{Q}^V & \tilde{Q}^A \\ \tilde{Q}^R & \tilde{Q}^K \end{pmatrix},
\tag{33}
$$

satisfy the equations

$$
\begin{aligned}
(m^2 - \tilde{m}^2) + \tilde{Q}^R &= -\frac{\kappa}{\omega^2 - \tilde{m}^2 + \tilde{Q}^R} \\
&\approx \frac{\kappa}{\tilde{m}^2} + \frac{\kappa}{\tilde{m}^4}(\omega^2 + \tilde{Q}^R) + \frac{\kappa}{\tilde{m}^6}\left[\tilde{Q}^R\right]^2, \\
2\pi i q_{\text{EA}}\delta(\omega) + \tilde{Q}^K &= \kappa \frac{2\pi i \left(q_{\text{EA}} + h^2\right)\delta(\omega) + \tilde{Q}^K}{|\omega^2 - \tilde{m}^2 + \tilde{Q}^R|^2}.
\end{aligned}
\tag{34}
$$

Causality of the spin response function requires $Q(t,t') \sim \theta(t-t')$. Furthermore, we have used that $\tilde{Q}^A(\omega)$ is the complex conjugate of $\tilde{Q}^R(\omega)$ and that $\tilde{Q}^V$ vanishes (hence the superscript) due to the normalization of the partition function $Z = 1$. Clearly, in the first equation, the linear term in $\tilde{Q}^R$ disappears for $\tilde{m}^4 = \kappa$, independent of $h$ (because $u = 0$ here). We conclude

$$\tilde{Q}^R(\omega) = -\sqrt{-\sqrt{\kappa}\left[(\omega + i0^+)^2 - r\right]}, \tag{35}$$

which is causal in the paramagnetic phase and gives rise to a phase transition when $r = m^2 - 2\sqrt{\kappa}$ vanishes at $\kappa = m^4/4$.

The second equation evaluated at $\omega = 0$ fixes the order parameter (we demand that $\tilde{Q}^K$ is a continuous function). Multiplying both sides with $G^R G^A$, inserting the retarded and advanced $\tilde{Q}^{R/A}$ and keeping only the leading term in $r$ one finds

$$q_{\text{EA}} = \frac{h^2 \kappa^{1/4}}{2\sqrt{r}}. \tag{36}$$

As expected, in the paramagnetic phase, the magnetization is linear in the longitudinal field $m \sim S \sim h$ and the Hubbard-Stratonovich field $Q \sim S^2$ is proportional to $h^2$. Furthermore, at the critical point, the system is gapless and has a divergent response to the external field, signified by $q_{\text{EA}} \sim r^{-1/2}$.

Below we obtain the Landau action by expanding in the small field $\tilde{Q}$ near the critical point. The above ensures that there will be no contribution $\sim Q^2 = \int_{t'} Q(t_1, t') Q(t', t_2)$ to the Landau action.

In the following, we will exclusively work with $\tilde{Q}$, $\tilde{G}_0$, and $\tilde{m}$ and therefore drop the tilde from here on.

## 3.2 Paramagnetic phase

Having established the proper order parameter field, we can now expand the action in the soft constraint $s_g$. For the discussion of the paramagnetic phase, an expansion to first order in $g$ is enough to obtain stable results known from equilibrium theory. On the other hand, an expansion to second order in $g$ is necessary to recover the spin glass phase [75].

Following the discussion above, we expand in small fields $Q$ to find the unconstrained action

$$is_0[Q] = -\frac{1}{2\kappa} \int_{t,t'} \left(\partial_t^2 + r\right) \text{tr}\left(\sigma^1 Q(t,t')\big|_{t=t'}\right) + \frac{i}{2\kappa} \int_{t,t'} h^\top(t) Q(t,t') h(t') - \frac{1}{6\kappa^{3/2}} \text{Tr}\left[(\sigma^1 Q)^3\right]. \tag{37}$$

To first order in $g$, the constraint on the spin length contributes a Hartree term

$$\begin{aligned} i\Delta s_g[Q] &= \frac{3ig}{2} \int_t \left(G^K + G^V\right)(t,t)\left(G^R + G^A\right)(t,t) \\ &\approx -\frac{3ig}{2\kappa^2} \int_t \left(Q^K + Q^V\right)(t,t)\left(Q^R + Q^A\right)(t,t), \end{aligned} \tag{38}$$

to the Landau action, which then reads

$$s[Q] = s_0[Q] + \Delta s_g[Q]. \tag{39}$$

To highlight the temporal structure of this action, it is useful to consider its diagrammatic representation shown in Fig. 3. We observe that the disorder gives rise to a term $\sim Q^3$ that relates the order-parameter fields at different times. As we will see below, it corresponds to a memory that for sufficiently large $\kappa$ causes the order parameter to become stiff, thereby excluding its relaxation at large relative times that is characteristic of the paramagnetic phase.

$$s[Q] = \bigcirc\!\!\Box + \frac{1}{2\kappa}\, \times\!\!-\!\!-\!\!\times + \frac{1}{6}\,\bigcirc\!\!\!\!\bigcirc + \frac{3}{2}\,\infty$$

Figure 3: Diagrammatic representation of the Landau action (39) at linear order in the soft-spin constraint $g$. For simplicity, we have suppressed the Keldysh structure. The inverse bare spin propagator for $Q = 0$ reading $1/(2\kappa)\sigma^1\delta(t_1 - t_2)(r + \partial_{t_2}^2)$ is depicted as open rectangle. $Q$ is shown as a straight line, $h$ is represented by a cross, the vertex $\frac{ig}{2\kappa^2}$ as a dot, and $G_0$ as open circle.

To find the critical disorder strength where the paramagnet freezes, we consider the equations of motion for $Q$, also known as Kadanoff-Baym equations [77,78], which are obtained from the saddle point condition

$$0 \stackrel{!}{=} \frac{\delta i s[Q]}{\delta Q(t_1, t_2)} \approx -\frac{1}{2\kappa}\sigma^1\delta(t_1 - t_2)\big(r + \partial_{t_2}\big) + \frac{1}{2\kappa^{3/2}}\int_t \sigma^1 Q(t_1, t)\sigma^1 Q(t, t_2)\sigma^1$$
$$+ \frac{i}{2\kappa}h(t_1)h^\top(t_2) + \frac{\delta i \Delta s_g[Q]}{\delta Q(t_1, t_2)}, \tag{40}$$

with

$$\frac{\delta i \Delta s_g[Q]}{\delta Q_{\bar\eta\bar\rho}(t_1, t_2)} = -\frac{3ig}{2\kappa}Q^K(t_1, t_1)\delta(t_1 - t_2)\delta_{\bar\eta\rho}, \tag{41}$$

where $\bar\eta$ denotes the opposite of $\eta$.

Following the general procedure outline in Sec. 2.2, we split the order parameter field into fast and slow components $Q = Q_f + Q_s$, where the evolution of $Q_s$ slows down indefinitely for $T \to \infty$. Looking for a paramagnetic solution, we require $Q_s = 0$ and make a time-translation invariant ansatz $Q(t, t') = Q(t - t')$. The equations of motion (40) therefore become diagonal in frequency space. Due to the absence of scattering at the current level of approximation, there is only one non-trivial equation of motion. Expanding for $h = 0$ in small frequencies $\omega$, one finds

$$\omega^2 + \frac{1}{\kappa^{1/2}}\big[Q^R\big]^2(\omega) = r - \frac{3ig}{\kappa}\int_\nu Q^K(\nu), \tag{42}$$

which has the thermal paramagnetic solution

$$Q^R(\omega) = -\kappa^{1/4}\sqrt{\Delta^2 - (\omega + i0^+)^2}, \tag{43}$$
$$Q^K(\omega) = 2\kappa^{1/4}\coth\frac{\beta|\omega|}{2}\sqrt{\Delta^2 - \omega^2}\,\theta\,(|\omega| - |\Delta|),$$

with the shifted mass $\Delta^2 = r - \frac{3ig}{\kappa}\int_\nu Q^K(\nu)$.

This reproduces the form of results from the analytically continued replica theory for $h = 0$ (up to relabelling of coefficients) developed in [75] and in [49] in the Keldysh framework. In particular, for small $g$ we reach a critical point $\Delta(r_c 0) = 0$, with

$$r_{c0} = -\frac{6g}{\kappa^{3/4}}\int_\omega \omega\coth\frac{\beta\omega}{2} \xrightarrow{\beta\to\infty} \frac{6g}{\kappa^{3/4}}\int_\omega |\omega|, \tag{44}$$

as well as $Q(\omega = 0) = 0$, which verifies the assumption that the shifted $Q$ is an order parameter for the Landau theory. After crossing the phase transition, we expect that $\Delta$ remains pinned to zero and that this can be achieved by introducing an Edwards-Anderson order parameter into the occupation function component, $Q_{EA}^K(\omega) = Q^K(\omega) + 2\pi i q^{EA}\delta(\omega)$, $q^{EA} > 0$. Indeed, inserting this ansatz into the equation of motion (42) we reproduce the known results [49,50]

$$
\begin{aligned}
Q^R(\omega) &= i\kappa^{1/4}\omega\,, \\
q^{EA} &= i\int_\nu Q^K(\nu)\Big|_{\Delta=0} - \frac{\kappa}{3g}r = \frac{\kappa}{3g}(r_{c0} - r)\,, \\
Q^K(\omega) &= 2i\kappa^{1/4}\omega\coth\frac{\beta\omega}{2}\,.
\end{aligned}
\tag{45}
$$

In particular, there is a gapless, damped mode.

## 3.3 Landau action to order $g^2$

The discussion of the spin glass phase requires a more careful discussion of the memory terms. In particular, beyond the critical point the disorder term $\sim Q^3$ renders the Landau action in Eq. (39) unstable. It is therefore necessary to continue the perturbative expansion in the soft-spin constraint $g$ to second order. As is shown in Fig. 4, there is only one term in the effective action that is of order $g^2$ and two-particle irreducible. It involves time-non-local fields and thus gives rise to memory effects that are essential for the stability of a spin glass. All other diagrams $\sim g^2$ are either disconnected or not two-particle irreducible and thus constitute at most a quantitative correction to the Hartree shift already discussed in the previous section. We therefore exclusively focus on the memory term at this order

$$
\begin{aligned}
i\Delta s_{g^2}[Q] = -\frac{3g^2}{4\kappa^4}\int_{t,t'}\Big[&(\mathrm{tr}(Q(t,t')Q(t',t))\mathrm{tr}(Q(t,t')\sigma^1 Q(t',t)\sigma^1) \\
&+\mathrm{tr}(Q(t,t')Q(t',t)\sigma^1)\mathrm{tr}(Q(t,t')\sigma^1 Q(t',t))\Big]\,.
\end{aligned}
\tag{46}
$$

Terms of this form are known as the primary cause of relaxation and thermalization in quench dynamics, see for example [79]. For the stability of the spin-glass it is therefore important to investigate the competition between the terms $\sim g^2$ that favor ergodicity and the disorder term $\sim Q^3$ that favors freezing.

Expanding the trace-log as before, we find the Landau action of the Sherrington-Kirkpatrick model with longitudinal and transversal fields

$$
s[Q] = s_0[Q] + \Delta s_g[Q] + \Delta s_{g^2}[Q]\,.
\tag{47}
$$

This action is the dynamical equivalent of the result recently reported in Ref. [50].

## 3.4 Asymptotic solution in the glass phase

In the previous section, we have found the Keldysh action corresponding to the equilibrium Landau action in replica theory. We will now consider the limit of late times and apply the general results of Sec. 2 to show how full replica symmetry breaking is recovered in the time domain.

In the limit of late times $T = (t_1 + t_2)/2 \to \infty$ the forward evolution scale drops out. This does exclude the spontaneous breaking of time translation invariance globally. Time translation invariance can, however, be broken in a scale-dependent way, as suggested by the reparametrization invariance of the aging action [6].

$$i\Delta s_{g^2}[Q] = \frac{1}{2}\left(90\times \phantom{xx} \right.$$

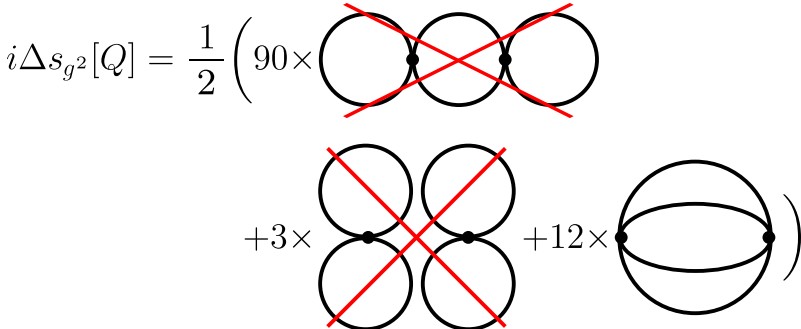

$$\left.+3\times \phantom{xx} +12\times \phantom{xx}\right)$$

Figure 4: Diagrammatic representation of the second order contribution of the soft-spin constraint $g$ to the effective action. The first diagram is not one-particle irreducible and corresponds to quantitative corrections to the tadpole diagram in Fig. 3. The second diagram is disconnected and therefore cancels against the normalization of the partition function. Consequently, we only retain the last contribution, which involves time-non-local fields and thus introduces a memory to the equation of motion that competes with the disorder term in the spin glass phase.

We thus bring the action into a form in which time translation invariance is used: $Q(t_1, t_2) = Q(t = t_1 - t_2)$. To this extent, one performs a Wigner expansion of the action (47) and drops all derivatives $\partial_T$. In all terms of the action the length of the time domain, $T$, factors out

$$is[Q]/T = -\frac{1}{2\kappa}\left(\partial_t^2 + r\right)\mathrm{tr}[\sigma^1 Q(t)]\Big|_{t=0} + \frac{ih_c^2}{2\kappa}\int_t Q^V(t) \tag{48}$$

$$+ \frac{1}{6\kappa^{3/2}}\int_{t,t'}\mathrm{tr}\left[Q(t)\sigma^1 Q(t')Q^\top(t+t')\right] + \frac{3ig}{2\kappa^2}\left[\mathrm{tr}(Q(t=0))\mathrm{tr}(\sigma^1 Q(t=0))\right]$$

$$-\frac{3g^2}{4\kappa^4}\int_t\left[\mathrm{tr}(Q(t)Q^\top(t))\mathrm{tr}(Q(t)\sigma^1 Q^\top(t)\sigma^1) + \mathrm{tr}(Q(t)Q^\top(t)\sigma^1)\mathrm{tr}(Q(t)\sigma^1 Q^\top(t))\right].$$

Following the procedure of Sec. 2, we split the field $Q$ in a slow and a fast component and similarly divide the action into a 'spin glass' part $s_{sg}$ that involves the slow field and a quantum part $s_q$ that describes the equilibration at short relative times. Since $Q_f(t)$ approaches zero for large arguments, we require $q_{EA} = -iQ_s^K(t=0)$. In analogy to the paramagnetic phase, in $s_q$ the terms $\sim g^2$ are not important at small frequencies, so we will neglect these by writing $s_q = s_{q,0} + \mathcal{O}(g^2)$. Since in the following, we will mostly concern ourselves with the slow field, we will drop its index from now on and simply refer to it as $Q$ (i.e. $Q \equiv Q_s$). One then has

$$s[Q] \approx s_{sg}[Q] + s_{q,0}[Q],$$

$$is_{sg}[Q]/T = -\int_t \mathrm{tr}[R_1 Q(t)Q^\top(t)\sigma^1] + \frac{ih_c^2}{2\kappa}\int_t Q^V(t) + \frac{R_2}{3}\int_{t,t'}\mathrm{tr}[Q(t)\sigma^1 Q(t')Q^\top(t+t')]$$

$$-\frac{R_3}{3}\int_t\left\{\mathrm{tr}[Q(t)Q^\top(t)]\mathrm{tr}[\sigma^1 Q(t)\sigma^1 Q^\top(t)] + \mathrm{tr}[Q(t)Q^\top(t)\sigma^1]\mathrm{tr}[Q(t)\sigma^1 Q^\top(t)]\right\},$$

$$is_{q,0}[Q]/T = \frac{1}{2\kappa}\int_\omega (\omega^2 - r)\mathrm{tr}[\sigma^1 Q_f(\omega)] - \frac{ih_c^2}{2\kappa}\mathrm{tr}[Q_f(\omega = 0)] \tag{49}$$

$$+ \frac{R_2}{3}\int_\omega \mathrm{tr}[\sigma^1 Q_f(\omega)\sigma^1 Q_f(\omega)\sigma^1 Q_f(\omega)] + iR_2 q_{EA}Q_{f\,11}(\omega = 0)\mathrm{tr}[\sigma^1 Q_f(\omega = 0)]$$

$$+ \frac{3ig}{2\kappa^2}\int_{\omega,\omega'}\mathrm{tr}[Q_f(\omega) + 2\pi\delta(\omega)Q_{EA}]\mathrm{tr}[\sigma^1(Q_f(\omega') + 2\pi\delta(\omega')Q_{EA})],$$

with

$$R_1 = -\frac{1}{2\kappa^{3/2}}\sigma^1 Q_f(\omega = 0)\sigma^1 \equiv \begin{pmatrix} R_1^K & R_1^R \\ R_1^A & R_1^V \end{pmatrix}, \qquad R_2 = \frac{1}{2\kappa^{3/2}}, \qquad R_3 = \frac{9g^2}{4\kappa^4}, \qquad (50)$$

where $R_1^A = R_1^R$. The saddlepoint of $s_{sg}$ and $s_{q,0}$ with respect to $Q$ and $Q_f$ respectively gives the coupled dynamical equations of the aging and ergodic components. Since we are looking for the qualitative form of the slow component $Q$, it is enough to consider $\delta s_{sg}[Q]/\delta Q$, which gives

$$0 = 2R_1^R Q^R(t) - R_2 \int_{t'} Q^R(t')Q^R(t-t') + \frac{2R_3}{3}\left[Q^{R2}(t) + 3Q^{K2}(t)\right]Q^R(t),$$

$$0 = R_1^K\left[Q^R(t) + Q^A(t)\right] + 2R_1^R Q^K(t) - i\frac{h^2}{2\kappa}$$

$$- R_2\left(\int_0^{T+t/2} dt' Q^K(t-t')Q^R(t') + \int_{t/2-T}^0 dt' Q^K(t-t')Q^A(t')\right) \qquad (51)$$

$$+ \frac{2R_3}{3}\left[Q^{K2}(t) + 3\left(Q^{R2}(t) + Q^{A2}(t)\right)\right]Q^K(t),$$

where we have kept the integration boundaries explicit, even though we have not done so before. The reason is, that, although we expect that the integration boundaries are irrelevant when we send $T \to \infty$, we want to show so explicitly in the following.

As is the case for the replica formulation, Eq. (51) has a ferromagnetic solution for which $Q^K(t)$ is a non-vanishing constant, while $Q^R(t) = 0$. In equilibrium, it can be shown that this solution is thermodynamically unstable [80], with the exact solution instead given by the Parisi function with full replica symmetry breaking [81, 82]. When considering quench dynamics on the other hand, one has to fix a boundary condition for $Q$ at large values of $|t|$. The difference between a system that exhibits aging and more conventional spontaneous symmetry breaking has to be encoded in the time scale on which the order parameter $q_{\mathrm{EA}}$ recovers from the perturbation at the boundary. Here, we will only discuss the equivalent of the spin glass solution, not the (im)possibility of a ferromagnetic phase.

Following the discussion of Kurchan [83], we expect $Q(t)$ to vary increasingly slowly as $t$ grows. In fact, each value of $Q^K(t)$ corresponds to a time scale on which the system thermalizes to an effective inverse temperature $X(t)$. This time scale is much longer than those of all previous (larger) values of $Q^K(t' < t)$. We can exploit this to simplify e.g. integrals of the form $\int_0^t dt' Q(t')Q(t-t')$. Specifically, for all values of $t'$ on the scale of $t$ one has $Q^K(t') = Q^K(t)$, while for $t-t' \sim t$ one has $Q^K(t-t') = Q^K(t)$. In other terms, the correlation $Q^K$ is ultrametric. At the same time, the generalized thermal response function $Q^R(t) = iX(t)\partial_t Q^K(t)$ vanishes much more quickly than $Q^K(t)$. This justifies the classical approximation of the general scheme in Sec. 2. In Eqs. (51) we therefore only keep the memory terms $\sim R_3$ with the highest power in $Q^K$ and drop the term proportional to $R_1^K$. At late times, the evolution thus is determined by a Martin-Siggia-Rose action [84, 85] corresponding to classical stochastic evolution.

Following these preparations, the second equation in (51) only involves time-local or Hadamard products of Keldysh Green's functions as well as the Keldysh component of causal convolutions. Both have been discussed in Sec. 2. Applying the partial integration Eq. (16), we find

$$0 = -2R_1^R q(u) + \frac{h^2}{2\kappa} + \frac{2R_3}{3}q(u)^3 - R_2\beta\left(2q(u)\int_g^1 dv\, q(v) + \int_0^u dv\, q(v)^2 + uq(u)^2 - 2q_{\mathrm{EA}}q(u)\right). \qquad (52)$$

Since $X(t = 0) = \beta/2$ is fixed by the temperature of the equilibrated part, we have parametrized $X(t) = \beta u/2$ with $u \in [0,1]$ and $Q^K(X(t)) = -iq(u)$. We thereby exactly recover the replica result [50].[1] Consequently, the Keldysh structure for $T \to \infty$ derives the rules of the replica limit. Physically, we can say that replica symmetry breaking corresponds to the inability of the system to fully thermalize to a single global inverse temperature $\beta$, even at arbitrarily late times/in the steady state. The assumption of the replica off-diagonal being independent of Matsubara frequency (hence including only the zeroth Matsubara frequency) is equivalent to the classical limit involving only the time-local generalized FDR (12).

Exploiting the analogy to the known solution from replica theory, it is easy to show that

$$q(u) = \begin{cases} q_h = \frac{1}{2}\left(\frac{3}{\kappa R_3} h^2\right)^{1/3}, & u \leq \frac{q_h}{q_{\text{EA}}} x, \\ q_{\text{EA}} \frac{u}{x}, & \frac{q_h}{q_{\text{EA}}} x < u < x, \\ q_{\text{EA}}, & x \leq u, \end{cases} \tag{53}$$

with $q_{\text{EA}}^2 = R_1^R/R_3$ and $x = \frac{2R_3 q_{\text{EA}}}{R_2 \beta} > 0$. Consequently,

$$X(q) = \begin{cases} 0, & q < q_h, \\ \frac{R_3}{R_2} q, & q_h < q < q_{\text{EA}}, \\ \frac{\beta}{2}, & q = q_{\text{EA}}, \end{cases} \tag{54}$$

which is consistent with the solution of Ref. [6]. We point out that $x > 0$ requires $Q^R(\omega = 0) > 0$, which, as we saw in Sec. 3.2, requires the disorder strength to exceed the critical value $\kappa > \kappa_c = m^4/4$.

What is left is to show that this is also a solution to the classical limit of the first equation in (51). This can be seen by integrating that equation with respect to $t$ and exploiting that with $Q^K(t)$ also $X(t)$ is an ultrametric function. One then finds

$$0 = 2R_1 \int_{q_{\text{EA}}}^q dq' X(q') + R_2 \left(\int_{q_{\text{EA}}}^q dq' X(q')\right)^2 - 2R_3 \int_{q_{\text{EA}}}^q dq' q'^2 X(q'), \tag{55}$$

which is indeed solved by (54).

# 4 Application: The quantum spherical $p$-spin model

Our second application is the quantum spherical $p$-spin model. We begin with a brief derivation of its effective action in the Keldysh formalism. We then apply the generalized thermal ansatz to the ultrametric aging component of the spin correlations. This procedure is then shown to reproduce the results known from replica formalism.

The Hamiltonian of the random $p$-spin model was first introduced in Ref [86] with the Ising counterpart discussed in Ref [87]. Its soft spin version was introduced shortly thereafter [25]. Upon inclusion of a spherical constraint, this classical model is known to exhibit 1-step replica symmetry breaking in thermal equilibrium [88]. Additionally, the transition between the paramagnetic and glass phase changes from second to first order continuous to discontinuous at low temperatures. There, the dynamical equations of motion predict a higher critical field strength than the equilibrium theory [39]. The same behavior is found for the quantum model, where the discontinuous transition is of first order also in the thermodynamic sense [89]. Due to the discrepancy between the dynamical and the equilibrium theory, it poses a critical test to the general arguments of Sec. 2.

---

[1]The only difference between their result and ours is the addition of $2\beta q_{\text{EA}} q(u)$ in Eq. (52). This is a consequence of the replica diagonal of the Parisi matrix being removed. It exactly compensates for the difference in the definition of $R_1$

## 4.1 Effective action

Technically, the discussion here follows closely that of Ref. [50], with modifications owed to the doubling of the time contour in the Keldysh approach. The spherical $p$-spin model is given by the Hamiltonian

$$H_{\text{int}} = \sum_{1 \leq i_1 < 1_2 < \cdots < i_p \leq N} J_{i_1 i_2 \ldots i_p} Z_{i_1} Z_{i_2} \ldots Z_{i_p}, \tag{56}$$

with Ising spins $Z_i = \pm 1$, $p \geq 3$ and the global spherical constraint $\sum_{i=1}^{N} Z_i^2 = N$. As for the Sherrington-Kirkpatrick model, we allow for longitudinal and transverse fields to couple to the spins (but neglect all commutators). The coupling constants $J_{i_1 i_2 \ldots i_p}$ are chosen randomly with a Gaussian distribution

$$\mathcal{P}(J_{i_1 \ldots i_p}) \propto \exp\left(-\frac{N^{p-1}}{p!} \frac{J_{i_1 \ldots i_p}}{J^2}\right). \tag{57}$$

Averaging the spins over some small regions, the Keldysh partition function,

$$Z = \int \mathcal{D}J_{i_1 \ldots i_p} \mathcal{P}(J_{i_1 \ldots i_p}) \int \mathcal{D}S \, e^{is[S]}, \tag{58}$$

can be written in terms of the continuous bosonic variable $S_{\eta,i}$, where the Latin index indicates the lattice site and the Greek index $\eta \in \{+, -\}$ denotes the branch of the Keldysh contour (see for example [76]). Due to the transverse field, the averaged spins obtain a massive dispersion. Hence, we can write the action as

$$\begin{aligned}
s[S] =& s_0[S] + s_h[S] + s_\kappa[S], \\
s_0[S] =& -\frac{1}{2} \int_t \sum_{\eta,i} \eta S_{\eta,i} \left(\partial_t^2 + m^2\right) S_{\eta,i}(t), \\
s_h[S] =& \int_t \sum_{\eta,i} \eta h_{\eta,i}(t) S_{\eta,i}(t), \\
s_\kappa[S] =& -i \int_t \sum_{\eta} \sum_{1 \leq i_1 < \cdots < i_p \leq N} \eta J_{i_1 \ldots i_p} S_{\eta,i_1} \ldots S_{\eta,i_p},
\end{aligned} \tag{59}$$

where the second term describes the coupling to the longitudinal external field and $s_\kappa$ accounts for the effect of the disorder Hamiltonian $H_{\text{int}}$.

Averaging over the Gaussian distribution of the coupling constants $J_{i_1 \ldots i_p}$ the disorder term is simplified to

$$\begin{aligned}
s_\kappa[S] =& \int_{t,t'} \frac{iJ^2}{p! N^{p-1}} \sum_{i_1 < \cdots < i_p} \left(\sum_\eta \eta S_{\eta,i_1} \ldots S_{\eta,i_p}\right)^2 \\
=& \frac{i\kappa}{4} \int_{t,t'} \frac{1}{N^{p-1}} \sum_{\eta\mu} \eta\mu \left(\sum_{i=1}^{N} S_{\eta,i}(t) S_{\mu,i}(t')\right)^p,
\end{aligned} \tag{60}$$

with $\kappa = J^2$. The global spherical constraint can be included using an auxiliary field $z_\eta(t)$ as

$$Z = \int \mathcal{D}S\mathcal{D}z \, e^{is[S,z]}, \tag{61}$$

with

$$s[S,z] = s[S] + \int_t \sum_\eta \eta z_\eta \left(S_{\eta,i}^2 - N\right). \tag{62}$$

At this point, the action has become purely local in the site index $i$. Without loss of generality, we may thus focus only on a single site, dropping the irrelevant site index.

Next, we introduce the bilocal field $\tilde{Q}_{\eta\mu}(t,t')$ as

$$\begin{aligned}
1 &= \int \mathcal{D}\tilde{Q}\, \delta\left[\tilde{Q}_{\eta\mu}(t,t') - S_\eta(t)S_\mu(t')\right] \\
&= \int \mathcal{D}\tilde{Q}\mathcal{D}\lambda \exp\left(\frac{i}{2}\int_{t,t'} \sum_{\eta\mu} \lambda_{\eta\mu}(t,t')\left[\tilde{Q}_{\eta\mu}(t,t') - S_\eta(t)S_\mu(t')\right]\right),
\end{aligned} \tag{63}$$

such that the disorder term becomes

$$s_\kappa[\tilde{Q}] = \frac{i\kappa}{4}\int_{t,t'} \sum_{\eta\mu} \eta\mu\, \tilde{Q}_{\eta\mu}^p(t,t'). \tag{64}$$

We can then perform the Gaussian integral over the averaged spin fields $S$, which gives

$$\begin{aligned}
Z &= \int \mathcal{D}\tilde{Q}\mathcal{D}\lambda\mathcal{D}z\, e^{is[\tilde{Q},\lambda,z]}, \\
s[\tilde{Q},\lambda,z] &= \frac{1}{2}\int_{t,t'} \sum_{\eta\mu} \eta\mu\, h_\eta(t)G_{\eta\mu}(t,t')h_\mu(t') - \int_t \sum_\eta \eta z_\eta(t) + \frac{1}{2}\mathrm{Tr}\left(\lambda\tilde{Q}\right) \\
&\quad + \frac{i\kappa}{4}\int_{t,t'} \sum_{\eta\mu} \eta\mu\, \tilde{Q}_{\eta\mu}^p(t,t') - \frac{i}{2}\mathrm{Tr}\ln(G),
\end{aligned} \tag{65}$$

where the trace is performed over time and the contour index alike, and we have introduced the inverse spin propagator

$$G^{-1}(t,t') = \delta(t-t')\left[-\left(\partial_t^2 + m^2\right)\sigma^3 + 2\mathrm{diag}(z_+, -z_-)\right] - \begin{pmatrix} \lambda_{11} & \lambda_{12} \\ \lambda_{21} & \lambda_{22} \end{pmatrix}(t,t'). \tag{66}$$

We now turn our attention to the saddle point equations of the action $s[\tilde{Q},\lambda,z]$. As these are most conveniently written in the $R/A/K$ basis, we introduce $z_{c/q} = z_+ \pm z_-$ such that in the new basis

$$G^{-1}(t,t') = \delta(t-t')\left[\left(-\partial_t^2 - m^2 + z_c(t)\right)\sigma^1 + z_q(t)\mathbb{1}\right] - \begin{pmatrix} \lambda^V & \lambda^A \\ \lambda^R & \lambda^K \end{pmatrix}(t,t'). \tag{67}$$

## 4.2 Late-time solution

We assume a constant longitudinal field $h_c = h = (h_+ + h_-)/2$, use that at the saddle point quantum fields vanish, and remember that $G^R(t,t) + G^A(t,t) = 0$ to write the saddle point

equations

$$0 \overset{!}{=} \frac{\delta s}{\delta z_q(t)} = -1 + \frac{i}{2}G^K(t,t) + \frac{h^2}{2}\int_{t',t''} G^R(t',t)G^A(t,t''),$$

$$0 \overset{!}{=} \frac{\delta s}{\delta z_c(t)} = 0,$$

$$0 \overset{!}{=} \frac{\delta s}{\delta \tilde{Q}^{R/K}(t,t')} = \frac{1}{2}\lambda^{R/K}(t,t') + \frac{i\kappa}{4}p\left[\tilde{Q}^{p-1}\right]^{R/K}(t,t'), \tag{68}$$

$$0 \overset{!}{=} \frac{\delta s}{\delta \lambda^A(t,t')} = \frac{1}{2}\tilde{Q}^R(t,t') - \frac{i}{2}G^R(t,t'),$$

$$0 \overset{!}{=} \frac{\delta s}{\delta \lambda^V(t,t')} = \frac{1}{2}\tilde{Q}^K(t,t') - \frac{i}{2}G^K(t,t') - \frac{h^2}{2}\int_{t'',t'''} G^R(t'',t)G^A(t',t'').$$

Here $\left[\tilde{Q}^p\right]^{R/K}$ refers to the retarded/Keldysh component of the $p$-th power of the matrix $\tilde{Q}$. These equations are to be compared with Eq. (3.17) in Ref. [50].

To simplify these equations even further, we specify $p = 3$. Furthermore, we introduce the real fields $Q^R(t,t') = i\tilde{Q}^R(t,t')$ and $Q^K(t,t') = \tilde{Q}^K(t,t')$, which then satisfy

$$Q^K(t,t) = 2,$$

$$Q^R(t,t') = \left[\delta(t-t')\left(\partial_t^2 + m^2 - z_c(t)\right) - \Sigma^R(t,t')\right]^{-1}, \tag{69}$$

$$Q^K(t,t') = \int_{t'',t'''} Q^R(t,t'')\Sigma^K(t'',t''')Q^A(t''',t'),$$

with the self-energies

$$\Sigma^R(t,t') = 3\kappa\, Q^R(t,t')Q^K(t,t'),$$

$$\Sigma^K(t,t') = \frac{3\kappa}{2}\left(\left[Q^K\right]^2(t,t') - \left[Q^R\right]^2(t,t') - \left[Q^A\right]^2(t,t')\right) + h(t)h(t'), \tag{70}$$

which are both real. In addition, $\Sigma^K$ is non-negative.

Following the arguments of Sec. 2, we distinguish between fast and slow fields $Q_{f/s}(t)$ in the time-translation invariant ansatz $Q(t) = Q_f(t) + Q_s(t)$. We then once again make a generalized thermal ansatz $Q_s^R(t) = -X(t)\theta(t)\partial_t Q_s^K(t)$. Since the slow field varies on a time scale that diverges as $T \to \infty$ this implies that the retarded Green's function decays more quickly than the Keldysh component. Consequently, the Keldysh self-energy simplifies as follows

$$\Sigma_s^K(t) = \frac{3\kappa}{2}\left(\left[Q_s^K\right]^2(t) - \left[Q_s^R\right]^2(t) - \left[Q_s^A\right]^2(t)\right)$$

$$= \frac{3\kappa}{2}\left[Q_s^K\right]^2(t). \tag{71}$$

From this, it follows immediately that the self-energy satisfies the generalized fluctuation-dissipation relation $\Sigma_s^R(t) = -X(t)\partial_t \Sigma_s^K(t)$. Similarly, the most slowly decaying contribution to the Keldysh component $Q_s^K$ must involve $\Sigma_s^K$ such that we can write

$$Q_s^K = Q^R \circ \Sigma_s^K \circ Q^A. \tag{72}$$

It is now more convenient to rewrite the equations of motion of the slow field in the more conventional form

$$\left[Q_f^R\right]^{-1} \circ Q_s^R = \Sigma_s^R \circ Q^R, \tag{73}$$

$$\left[Q_f^R\right]^{-1} \circ Q_s^K = \Sigma_s^K \circ Q^A + \Sigma_s^R \circ Q_s^K. \tag{74}$$

In the case of dissipative dynamics, these equations coincide with those derived by Sompolinsky and Zippelius [11,41] and solved by Cugliandolo and Kurchan [5]. As has been noted before [25], we find that these equations of motion satisfied by the $p$-spin model are surprisingly similar to those derived from mode coupling theory in the context of structural glasses [26].

Assuming ultrametricity, we satisfy all conditions required for the general argument of Sec. 2, where we showed that the matrix multiplication in replica space is identical to the Keldysh component of the product of functions in Keldysh space. From the general matrix multiplication follows the same statement also for matrix inversion. Hence, we conclude that the equation for the Keldysh component of the expression

$$Q(t) = \left[\delta(t)\sigma^1(\partial_t^2 + m^2 - z_c) - \Sigma(t)\right]^{-1}, \tag{75}$$

or equivalently the solution to (74) is similar to that obtained in replica formalism (see for example Eq. (3.17) in Ref. [50], which differs in the conventions for mass and coupling strength).

In summary, we find

$$Q_s^K(u) = \begin{cases} q_0 = -\frac{q_f \sigma_0 (\sigma_f - 2z)}{(\sigma_f + x(\sigma_1 - \sigma_0) - 2z)^2}, & u < x, \\ q_1 = q_0 - \frac{q_f(\sigma_1 - \sigma_0)}{\sigma_f + x(\sigma_1 - \sigma_0) - 2z}, & u > x, \end{cases} \tag{76}$$

with the shorthand notation

$$\Sigma_s^K(u) = \frac{3\kappa}{2}\left[Q_s^K\right]^2(u) + h^2 = \begin{cases} \sigma_0, & u < x, \\ \sigma_1, & u > x. \end{cases} \tag{77}$$

Furthermore, the fast field satisfies

$$\begin{aligned} G_f^K(t) &= G_f^R \circ \Sigma_f^K \circ G_f^A, \\ \Sigma_f^K(t) &= \frac{3\kappa}{2}(Q_f^K(t) + 2q_1)Q_f^K(t), \end{aligned} \tag{78}$$

which we abbreviated above as $q_f = Q_f^K(t=0)$ and $\sigma_f = \Sigma_f^K(t=0)$. Finally, the Lagrange parameter $z = (z_c - m^2)/\beta$ is fixed by the additional constraint

$$Q^K(u=1) \equiv q_1 + q_f \equiv Q_s^K(t=0) + Q_f^K(t=0) \overset{!}{=} 2. \tag{79}$$

Conversely to Eq. (76), the effective inverse temperature mirrors the structure of 1-step RSB

$$X(q) = \begin{cases} 0, & q < q_0, \\ \frac{\beta x}{2}, & q_0 < q < q_1, \\ \frac{\beta}{2}, & q = q_1. \end{cases} \tag{80}$$

Note, that once again, it is not possible to reconstruct $Q_s^K(t)$ because the information on the time-dependence was lost during the change of variables $t \to X(t)$ in Eq. (16). Furthermore, the breakpoint $x$ has to be determined by an additional criterion, requiring either marginal stability or minimization of the free energy [39,50].

Due to the equivalence between the ultrametric Keldysh and the replica formalism, we conclude that our approach finds the same critical point and one-step RSB as reported in Ref. [50], provided the same condition for $x$ is used. On the other hand, at any finite time $T$,

ultrametric relations must be violated and an analysis similar to that of Ref. [26] shows that on a finite time interval in the one-time formulation, correlations and response functions of the spin glass phase are indistinguishable from those of a ferromagnet.

The comparison between ultrametric Keldysh and replica formalism for the $p$-spin model has already been addressed by Crisanti et al. some 31 years ago [39]. Although they use a slightly different ansatz for the generalized fluctuation-dissipation relation in the aging regime

$$Q^R(t) = -x\theta(t)\partial_t Q^K(t), \tag{81}$$

where $x \in [0, 1]$ corresponds to the position of the discontinuity in the replica formalism. In the case of 1-step RSB without a longitudinal field, this ansatz also reproduces the replica equations. The reported difference between the dynamical and equilibrium critical temperature is related in part to the different conditions used to fix $x$. This is consistent with the results previously reported in Ref. [10]. In the dynamical case, matching with the fast dynamics implies a marginal stability condition as opposed to a minimization of the free energy in equilibrium. Furthermore, as we had anticipated below Eq.(14) for models with a finite number of replica symmetry breaking steps, the Keldysh Green's function of the spherical $p$-spin model does not become ultrametric at late times [5]. Consequently, the aging dynamics of the spherical $p$-spin model never reaches thermal equilibrium.

An intuitive explanation of this observation can be given using the Thouless-Anderson-Palmer free energy [90]. One finds that the dynamics of the spherical $p$-spin model gets stuck in local minima that are separated from the equilibrium solution by energy barriers that diverge in the thermodynamic limit. For comparison, the slow evolution of the Sherrington-Kirkpatrick model is explained by an entropic effect: As the system relaxes, it evolves through a series of saddle points with an ever decreasing number of unstable directions resulting in long, but finite escape times [21].

# 5 Discussion

The results presented in this article rely on the existence of a finite temperature to which the system equilibrates on short relative times $t < \tau_{\text{erg}}$, see Fig. 1(b). Specifically, as we send the center-of-mass time $T \to \infty$, the ultrametric solutions (54)(80) are parametrized by the inverse temperature $\beta$. However, in a spin glass, no global equilibrium is reached. We identify the absence of a global temperature as the characteristic property of the ultrametric spin glass. This is independent of the breaking of time translation invariance at finite center-of-mass times $T$. We also address to which extent these conclusions apply to quantum critical quenches at zero temperature.

## 5.1 Spontaneous breaking of thermal symmetry

The non-analytic behavior of the ultrametric solution at $x$ emerges in the temporal thermodynamic limit $T \to \infty$ (in space, the mean-field system is assumed to be in the thermodynamic limit by construction). The ultrametric solution corresponds to a spontaneous breaking of the thermal (or Kubo-Martin-Schwinger, KMS) symmetry [91–94]

$$S_{\eta,i}(t) \to S_{\eta,i}(-t + i\eta\beta/2), \qquad i \to -i, \qquad h \to -h, \tag{82}$$

which is present in the stationary state of an ergodic system with a time-independent Hamiltonian generator of dynamics characterized by an inverse temperature $\beta$. Via our construction, replica symmetry breaking thus gets stringently tied to the spontaneous breaking of thermal symmetry – or more physically speaking, of ergodicity.

We emphasize that, since $T$ drops out of the equations of motion at asymptotically late times, which can be seen explicitly in Eq. (48), all microscopic details of the quench protocol disappear from the problem. The time-translation invariant discussion presented here is, therefore, independent of the details of the aging process. It instead extracts solely the universal property common to all classical glasses: The spontaneous breaking of thermal symmetry. The emergence of this broken symmetry at finite times was previously anticipated by Kurchan [83].

For glasses, it is found that a weak long-term memory is necessary to preclude thermalization on all scales. Although this implies that time translation symmetry remains broken at any finite time $T$ following a quench, our time translation invariant approach clarifies that the persistence of broken time translation invariance, and thus aging, should not be equated to ergodicity breaking in the stationary state. Instead, the emergence of reparametrization invariance lifts this connection at asymptotically late times [5,6]. In the absence of reparametrization invariance, the system spin glass can retain some information about the initial state as is the case for the mixed $p$-spin model [95]. It will be interesting to see to which extent this is recovered in the ultrametric Keldysh formalism.

## 5.2 Zero temperature limit

The finite temperature spin glasses discussed here are solved by the classical ansatz $G^R(t) \sim \beta \partial_t G^K(t)$ with different scaling dimensions for response and correlation functions. The classical scaling, therefore, requires the existence of a time scale that enters the asymptotic solution as inverse temperature. For a quench through the quantum critical point at zero temperature, one, therefore, expects one of two options: Either $\beta$ emerges as a result of the finite energy density imposed upon the system during the quench, or the absence of a fixed time scale suggests quantum scaling

$$G^R_s \sim G^K_s . \tag{83}$$

In the following, we will address the implications of quantum scaling. With Eq. (83), it is not possible to expand the equations of motion in powers of $G^R$. Furthermore, the failure of the generalized thermal ansatz indicates the necessity of a dynamic Parisi function.

The characteristic observable feature of a glass is aging, which implies that correlations $G^K_s(t)$ decay infinitely slowly as $T \to \infty$. In the quantum regime, assuming the above scaling, the same must apply to the response function $G^R_s$, and thus the self-energy $\Sigma^R_s$. Hence, as the infrared cutoff $T^{-1}$ is sent to zero, memory integrals of the form $\Sigma^R_s \circ G^K_s$ diverge. We emphasize the similarity of this argument to the Mermin-Wagner theorem that prevents spontaneous symmetry breaking due to infrared fluctuations – here, these fluctuations prevent the ergodicity breaking identified in the classical case above, upon removing the infrared cutoff $T \to \infty$. Consequently, the quantum regime characterized by Eq. (83) is always transient and bounded by the energy density imparted upon the system by the initial quench. According to this argument, at asymptotically late times, spin glasses are necessarily classical (see also [45,96]) with a temperature determined by the energy density after the quench.

We re-emphasize, however, that the argument here relies on the assumption of a common scaling of retarded and Keldysh Green's functions. This raises the question of whether more general forms of ergodicity breaking could be realized at zero temperature.

Recent experiments are performed at very low temperatures and finite times [60–65]. In addition to possible asymptotic symmetry-breaking phenomena, weak quenches at zero temperature could also display interesting intermediate-time dynamical phenomena related to their quantum mechanical microscopic physics.

At the current level of the analysis presented here, it is not possible to recover the time scales associated with the effective temperature $X$, which hinders the investigation of transient regimes. However, by continuing the Wigner expansion, it is possible to systematically restore corrections due to the boundary at $t = 2T$ and derivatives with respect to the center-of-mass time. It is then possible to work backward from the latest times to recover the explicit time dependence of the aging solution, including a potential transient quantum critical regime.

## 6 Outlook

Recent realizations of spin glasses with Rydberg atoms are affected by decoherence due to dephasing caused by fluctuations in the external fields (i.e. lasers) and spontaneous emission from the Rydberg state [61, 66]. Although typical decoherence rates are several orders of magnitude smaller than the interaction strength, such that aging dynamics are expected to be observable, they are relevant perturbations that a more realistic description of the system will have to take into account. This necessitates the treatment of an open system with a time evolution governed by the Lindblad equation

$$\partial_t \rho(t) = -i[H, \rho] + \kappa \sum_i \left( L_i \rho L_i^\dagger - \frac{1}{2}\{L_i L_i^\dagger, \rho\} \right). \tag{84}$$

Here, $\rho(t)$ denotes the density matrix, the Hamiltonian $H$ is that of Eq. (25), and the Hermitian Lindblad operators $L_i = \sigma_i^3$ describe dephasing noise that acts incoherently on all atoms. The decoherence introduced by the Lindblad operators causes heating. Specifically, for Hermitian $L_i$, the stationary state has infinite temperature. Dephasing, therefore, introduces a time scale beyond which the system becomes paramagnetic, independent of the initial quench. At late times, dephasing needs to be taken into account by simulations of the experimental systems.

It is a strength of the Keldysh field theory that the inclusion of decoherence is very natural and requires little additional effort [76]. This is in contrast to microscopic approaches like exact diagonalization or matrix product states, particularly in quantum systems at low temperatures, when the system becomes highly entangled [73]. Despite this advantage, simulations of the glass phase, even in mean-field models, remain challenging. The reason is the weak long-term memory, which precludes using a finite cutoff time for memory integrals. The numerical effort therefore scales with time to the third power, which currently limits this method to short times. However, these limitations can be lifted [97] and long-time simulations of the quench dynamics will be addressed in the future [98].

Finally, we mention the connection to Sachdev-Ye-Kitaev (SYK) models, which have quantum 'spin liquid' ground states [99]. These states are quite distinct from the spin glass ground states considered in the present paper, as they do not have any aging behavior, and are described by a replica diagonal saddle point. The low energy theory of SYK models exhibits an emergent time reparameterization symmetry while preserving thermal symmetry. This has enabled a detailed understanding of their quantum dynamics at a finite number of spins $N_s$, well beyond the $N_s = \infty$ saddle point. The quantum spin glass states considered in the present paper also have an emergent time reparameterization symmetry, but the glassy dynamics break thermal symmetry [100]. All our analysis here has been in the $N_s = \infty$ saddle point theory, and it would be interesting to adapt the SYK technology to understand the structure of the finite $N_s$ theory. However, the broken thermal symmetry makes this task considerably more difficult.

# Acknowledgments

We thank Alexander Altland, Giulio Biroli, Leticia Cugliandolo, Antoine Georges, Nikita Kavokine, Jorge Kurchan, Olivier Parcollet, Rhine Samajdar, Marco Schiró, and Maria Tikhanovskaya for valuable discussions. S.S. thanks Rhine Samajdar and Maria Tikhanovskaya for an earlier collaboration [50].

**Funding information** The work of J.L. and S.D. was supported by the Deutsche Forschungs-gemeinschaft (DFG, German Research Foundation) CRC 1238 project number 277146847. S.S. has been supported by the U.S. Department of Energy under Grant DE-SC0019030.

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
