# Peer review of "Replica symmetry breaking in spin glasses in the replica-free Keldysh formalism"

_SciPost Physics, doi:SciPost Phys. 17, 160 (2024)_

## Round 1 · Referee Report · Anonymous (Referee 1) · 2024-8-1

Strengths

1- General approach using the Keldysh formalism

Weaknesses

1- It does not consider already known results in the field. 2- Not easy for people not familiar with Keldysh formalism 3- Some definitions are missing 4- Notation not always clear

Report

The authors present a Keldysh formalism in which replica symmetry breaking in spin-glasses is analysed in terms of a dynamic theory. They show that ultrametricity emerges in the asymptotic limit of infinite long times and they study the slow aging dynamics in the reparametrization invariance regime of Cugliandolo and Kurchan. They apply their formalism to the  classical and quantum Sherrington-Kirkpatrick and spherical p-spin spin-glass models.

As a general comment on the manuscript it seem that the Authors are not aware of some results in the field. The emergence of ultrametricity in the asymptotic limit of infinite long times is not a new results, see e.g. Crisanti and Ritort J. Phys. A: Math. Gen. 36, R181 (2003) Sec. 4 and 6. The link between dynamics and statics (in mean-field models) is well established for almost 20 years. In particular, how to interpret r steps in replica symmetry breaking in terms of diverging (and ordered) time scales (we guess what they refer to as “ultrametric dynamics”?) has been addressed by Crisanti and Leuzzi in PRB 75 144301 (2007).

Beside this there are a number of technical comments on the manuscript and a couple of more general observations on some of the claims carried out in the work. We report them in the following in order of appearance in the manuscript.

Based on this we cannot recommend the manuscript for publication in the present form. The Authors must go through a substantial rewriting before the manuscript can be considered for publication to give credit to previous works and fit their results within the already known results.

Requested changes

1) Refs. 39 and 48 are the same reference. 

2) Lines 99-100 Concerning the related connection between supersymmetry and thermal symmetry in the paramagnetic phase of spin glasses that was previously found by Kurchan [48] an important follow up, linking the multi-scale dynamics to mode-coupling theory also beyond schematic theories for continuous variables was developed and presented in a series of papers in the 2010’s. See Caltagirone at al. PRL 108, 085702 (2012), Parisi, Rizzo PRE 87, 012101 (2013) and references therein. In particular, one can look at Ferrari et al. PRB 86 014204 (2012) as a specific application to p-spin models analysed in this manuscript (see also lines 601-603).

3) Line 145:  the relationship between distance (in the ultra metric space) and Parisi matrix overlap elements $P_{ab}$ is written as 1/P_{ab}. In this way a zero overlap corresponds to an infinite distance, i.e., an infinite edge of a triangle in the ultra metric space. Is this more conveniente than considering d_{ab} = 1 - P_{ab} (thus having limited distances \in [0,1])?

4) Section 2.1. 
About Parisi matrix manipulations the authors might consider or at least acknowledge the approach of  Replica Fourier Transform of Crisanti and De Dominicis, Nuclear Physics B, 891 (2015). Also what the Authors call “u” is usually indicated with “x”. To avoid possible confusion the Authors should use the now standard notation “x”. The letter “u” is also used un Sec. 3.1 with a complete different meaning.

5) Section 2.2. The dynamics of disorder systems was first addressed in the late 70’s in the seminal works of Martin, Siggia, Rose, de Dominicis, see eq. de Dominics Phys. Rev B19, 4913 (1978). This approach leads naturally to a dynamic theory with the same structure of Keldysh dynamics. The Authors should comment somewhere about how the Keldysh formalism for spin-glass dynamics is related to the DeDominicis-Peliti-Martin-Siggia-Rose dynamic mean-field theory. Here or in Sec. 4.1, or both.

6) Line 178 Please report after Eq. (6) the definition od G and \Sigma and their components. Also report what the apices K, R and A stand for.

7) Lines 181,186 “s” is both the spin variable and the “slow” subindex. 

8) I do not find a definition for G^A, see also comment 6).

9) Lines 189-192.  In the aging regime is b << 1. That is, b/\Lambda < 1/ Λ. 
In the fast regime b \to 1, i.e,. b/\Lambda \to 1/\Lambda. 
Is b  sent to 1 from left or right? The $b \to 1^-$ limit might seem reasonable to interpolate between slow and fast, and would be consistent with the decomposition of integral (8), but in this case it would never be  1/\Lambda \leq |t| \laq b/\Lambda. 
The Authors should clarify how this limit is taken and how the two times scales are conceived in terms of the regularizer b and of the cut-off Λ.

10) Lines 214-215. 
The Ansatz of CHS, Ref 35, initially put forward in the p-spin spherical model (i.e. 1RSB stable at low T) has been extended to a generic number of time scales in the context of the spherical 2+p spin model (displaying also spin-glass phases with a infinite number of breakings) in “Equilibrium dynamics of spin-glass systems” by Crisanti. Leuzzi PRB 75 144301 (2007). That extension solved the issue of the instability of Sompolinsky dynamics in the static limit. The author should comment whether Ansatz (10) is consistent with this.

Also later one, on Lines 298-299, the authors write “Despite several attempts at recovering the results of replica theory from the dynamical  equations at late times [30–33], discrepancies remain [48].” Here, as well, the above mentioned work “Equilibrium dynamics of spin-glass systems” should be considered, in which such discrepancies are overcome.  See, furthermore, lines 641-642 where it is written “Consequently, the aging dynamics of the spherical p-spin model never reaches equilibrium.”

Also, the replica field-theory approach to the dynamic critical slowing down might be interesting to be compared to. See e.g., the above mentioned Paris, Rizzo PRE 87, 012101 (2013)  as well as Leuzzi, Rizzo PRB 109, 174211 (2024). This stems from the structure of the superfield built in Ref. 39/48 by Kurchan.

11) Line 321, Eq (23). Please mention that the sigma’s are Pauli matrices. 

12) Section 3.1 Line 334-5. The authors write: “An important distinction between the new platforms and classical glasses is the finite lifetime of the excited states due to spontaneous emission”. How long is such lifetime? Is it long enough for aging to be experimentally detected?

13) Eq. (24). Please specify all the elements in the action: m, h_\sigma, etc.. 
The author might consider a Greek index different from \sigma that takes the values +,- to denote the branch of the contour.
Indeed, \sigma denote the Pauli matrices here.

14) Eq. (31).  What does the superscript V stand for?

15) Line 425 In Eq. (24), in order to describe the critical behavior the authors move from Pauli matrices to soft spins S, i.e., scalar commuting variables in a quartic potential. However, in s_\kappa the Pauli matrices stay and are traced over. Some easy going description might perhaps help the reader to follow the manuscript without compulsorily resorting to ref. 45.

16) Line 491-499. How is the ferromagnetic phase mentioned here characterized in terms of the correlation functions involved in Eq. (49)?  When talking about instability in this dynamic equations, where slow and fast decomposition is assumed, the authors cite the work of de Almeida-Thouless and the Parisi solution, i.e., the instability of the equilibrium solution. The authors should clarify better what they mean by instability in this dynamic case and how it can be analyzed in the dynamic Keldysh theory context, apart  from analogies with the statics.

17) Lines 518-520: the authors write “Physically, we can say that replica symmetry breaking corresponds to the inability of the system to fully thermalize to a single global inverse temperature β even at arbitrarily late times/the steady state.” Is the sentence complete? Are the authors trying to say that if a RSB occurs then  there is no single state equilibrium phase? Equivalently, that RSB signals multi-equilibria phases?

18) Lines 542-544 In Ref. [75] Derrida introduces the Hamiltonian of the random p-spin model with Ising spins and shows that in the limit p->\infty it is equivalent to the random energy model. This was further discussed by Gross and Mozart, Nucl. Phys. B240, 431 (1988). The soft spin version of the Ising p-spin model introduced by Derrida was introduced and discussed by Kirkpatrick and Thirumalai Ref[21]. The spherical p-spin spin glass model was first introduced in Ref. [77].

19) Line 545. The SG-PM transition in the spherical p-spin model is not a first order one in the thermodynamic sense: there is no latent heat and no phase coexistence. The order parameter has a discontinuous transition and a dynamic transition (Ref [35]) arises next to the static one. This transition has been termed Random First Order Transition by Wolynes and coworkers in the 80's, including Ref. [21].

The authors should comment somewhere about how the Keldysh formalism for spin-glass dynamics is related to the DeDominicis-Peliti-Martin-Siggia-Rose dynamic mean-field theory. For instance in Sec. 4.1.

20) Line 586.  “..and in the case of $\Sigma^K$ non-negative.” Sentence unclear, please complete.

21) Lines 627-628. The authors should clarify what it is meant by saying that the spin-glass phase is indistinguishable from a ferromagnet, in connection to mode coupling theory.
In glass forming liquids at the Mode Coupling temperature the correlation of density fluctuations does not decay to zero anymore but density fluctuations have, nevertheless, still zero mean. Here the analogy should be: non trivial non-zero spin-spin correlation at long times but zero magnetization.   Or it is something else occurring?

22) Lines 638-642 The connection between dynamics and statics has been discussed in Crisanti. Leuzzi PRB 75 144301 (2007) where it is shown that the CHS dynamics of Ref [35] generalized to any number of relaxation (diverging) times scale does lead asymptotically to the static RSB in the Parisi scheme. See also Crisanti de Dominics J. Phys A: Math. Theor. 44 115006 (2011) where it is shown that the static susceptibility in the Parisi RSB scheme coincides with the static limit of the CHS dynamics.

23) Line 721. About experimental realisations. What are the experimental time-scales in the experiments with dephasing? Is it practically feasible to probe the aging regime in those systems?

24) Eventually it is important to notice that reparametrization invariance is proved untrue as soon as the model is not “pure”, that is when more than a single p-spin interaction term is present in the spin-glass Hamiltonian. This is never mentioned throughout the manuscript where only the pure p-spin model is considered but is a crucial issue of critical slowing down and aging in spin-glass systems. See “Rethinking Mean-Field Glassy Dynamics and Its Relation with the Energy Landscape: The Surprising Case of the Spherical Mixed 𝑝-Spin Model" by Folena, Franz, Ricci-Tersenghi PRX 10, 031045 (2020)

25) In discussing the p-spin model the Authors should mention that within the 1RSB solution there is no ultrametricity.

Recommendation

Ask for major revision

  • validity: low
  • significance: ok
  • originality: low
  • clarity: low
  • formatting: acceptable
  • grammar: reasonable

Author:  Johannes Lang  on 2024-09-04  [id 4738]

(in reply to Report 1 on 2024-08-01)

Referee: The authors present a Keldysh formalism in which replica symmetry breaking in spin-glasses is analysed in terms of a dynamic theory. They show that ultrametricity emerges in the asymptotic limit of infinite long times and they study the slow aging dynamics in the reparametrization invariance regime of Cugliandolo and Kurchan. They apply their formalism to the classical and quantum Sherrington-Kirkpatrick and spherical p-spin spin-glass models. As a general comment on the manuscript it seem that the Authors are not aware of some results in the field. The emergence of ultrametricity in the asymptotic limit of infinite long times is not a new results, see e.g. Crisanti and Ritort J. Phys. A: Math. Gen. 36, R181 (2003) Sec. 4 and 6. The link between dynamics and statics (in mean-field models) is well established for almost 20 years. In particular, how to interpret r steps in replica symmetry breaking in terms of diverging (and ordered) time scales (we guess what they refer to as “ultrametric dynamics”?) has been addressed by Crisanti and Leuzzi in PRB 75 144301 (2007). Beside this there are a number of technical comments on the manuscript and a couple of more general observations on some of the claims carried out in the work. We report them in the following in order of appearance in the manuscript. Based on this we cannot recommend the manuscript for publication in the present form. The Authors must go through a substantial rewriting before the manuscript can be considered for publication to give credit to previous works and fit their results within the already known results.

Response: We are very grateful to the referee for a careful reading of the manuscript and the numerous suggestions for improvements. We have considered these carefully, and our responses are below, along with a description of the changes to our paper. We maintain that our paper has numerous advances which go beyond those in the literature. But, guided by the many helpful suggestions from the referee, we have now made the relationship between our results and the previous work clearer and more explicit. Specifically, our paper establishes a new and explicit connection between the algebra of Parisi matrices, and the Dyson-Keldysh equations of quantum systems. It also contains the first complete treatment of the dynamics of the infinite-range, quantum, Ising spin glass in a regime where the long-time limit has spin glass order with full replica symmetry breaking. We have now rewritten the abstract, and made significant changes to Sections 1 and 2 to make the novelty of our work and its connections to the earlier work clearer. We have also cited the two papers mentioned above, which are definitely relevant – it was an oversight to overlook them, and we thank the referee for informing us about these papers. Added: Crisanti and Ritort J. Phys. A: Math. Gen. 36, R181 (2003)

1) Refs. 39 and 48 are the same reference.

Response: This has been fixed.

2) Lines 99-100 Concerning the related connection between supersymmetry and thermal symmetry in the paramagnetic phase of spin glasses that was previously found by Kurchan [48] an important follow up, linking the multi-scale dynamics to mode-coupling theory also beyond schematic theories for continuous variables was developed and presented in a series of papers in the 2010’s. See Caltagirone at al. PRL 108, 085702 (2012), Parisi, Rizzo PRE 87, 012101 (2013) and references therein. In particular, one can look at Ferrari et al. PRB 86 014204 (2012) as a specific application to p-spin models analysed in this manuscript (see also lines 601-603).

Response: We thank the referee for pointing out these references, which have been added in the introduction to Sec. 2.

3) Line 145: the relationship between distance (in the ultra metric space) and Parisi matrix overlap elements Pab is written as 1/P_{ab}. In this way a zero overlap corresponds to an infinite distance, i.e., an infinite edge of a triangle in the ultra metric space. Is this more conveniente than considering d_{ab} = 1 - P_{ab} (thus having limited distances \in [0,1])?

Response: The choice 1/P_{ab} is arbitrary. The expression is not used any further and merely serves as an example for an allowed measure of distance. As compared to the alternative 1-P_{ab}, it has the slight advantage of not requiring P to be normalized to lie within the interval [0,1], which at this point of the paper has not been established yet. A comment concerning the freedom of choice has been added to the manuscript.

4) Section 2.1. 
About Parisi matrix manipulations the authors might consider or at least acknowledge the approach of Replica Fourier Transform of Crisanti and De Dominicis, Nuclear Physics B, 891 (2015). Also what the Authors call “u” is usually indicated with “x”. To avoid possible confusion the Authors should use the now standard notation “x”. The letter “u” is also used un Sec. 3.1 with a complete different meaning.

Response: A statement regarding the possibility of an equivalent formulation or the analogy between dynamical theory in frequency space and replica formalism in replica fourier space has been added at the end of Sec 2. To avoid any potential confusion, at the beginning of Sec. 2.1 we point out that sometimes u is called x in spin glass literature and have changed the interaction strength parameter from u to g in Sec. 3.

5) Section 2.2. The dynamics of disorder systems was first addressed in the late 70’s in the seminal works of Martin, Siggia, Rose, de Dominicis, see eq. de Dominics Phys. Rev B19, 4913 (1978). This approach leads naturally to a dynamic theory with the same structure of Keldysh dynamics. The Authors should comment somewhere about how the Keldysh formalism for spin-glass dynamics is related to the DeDominicis-Peliti-Martin-Siggia-Rose dynamic mean-field theory. Here or in Sec. 4.1, or both.

Response: We now comment explicitly in Sec. 3.4 on the limit of classical stochastic evolution corresponding to the Martin-Siggia-Rose action.

6) Line 178 Please report after Eq. (6) the definition od G and \Sigma and their components. Also report what the apices K, R and A stand for.

Response: The Retarded G^R is defined in Eq. (9), the Advanced component G^A in the first line underneath Eq. (9). These superscripts are standard and require no further comment. G^K is known as Keldysh component. This is now pointed out explicitly above Eq. (7),

7) Lines 181,186 “s” is both the spin variable and the “slow” subindex.

Response: There is no risk of confusion. The subscript “s” is exclusively reserved for “slow” variables.

8) I do not find a definition for G^A, see also comment 6).

Response: See the reply to 6).

9) Lines 189-192. In the aging regime is b << 1. That is, b/\Lambda < 1/ Λ. 
In the fast regime b \to 1, i.e,. b/\Lambda \to 1/\Lambda. 
Is b sent to 1 from left or right? The b→1− limit might seem reasonable to interpolate between slow and fast, and would be consistent with the decomposition of integral (8), but in this case it would never be 1/\Lambda \leq |t| \laq b/\Lambda. 
The Authors should clarify how this limit is taken and how the two times scales are conceived in terms of the regularizer b and of the cut-off Λ.

Response: We thank the referee for pointing out this typo. The correct order of conditions reads 1/\Lambda \leq |t| \leq b/\Lambda.

10) Lines 214-215. 
The Ansatz of CHS, Ref 35, initially put forward in the p-spin spherical model (i.e. 1RSB stable at low T) has been extended to a generic number of time scales in the context of the spherical 2+p spin model (displaying also spin-glass phases with a infinite number of breakings) in “Equilibrium dynamics of spin-glass systems” by Crisanti. Leuzzi PRB 75 144301 (2007). That extension solved the issue of the instability of Sompolinsky dynamics in the static limit. The author should comment whether Ansatz (10) is consistent with this.

Also later one, on Lines 298-299, the authors write “Despite several attempts at recovering the results of replica theory from the dynamical equations at late times [30–33], discrepancies remain [48].” Here, as well, the above mentioned work “Equilibrium dynamics of spin-glass systems” should be considered, in which such discrepancies are overcome. See, furthermore, lines 641-642 where it is written “Consequently, the aging dynamics of the spherical p-spin model never reaches equilibrium.”

Also, the replica field-theory approach to the dynamic critical slowing down might be interesting to be compared to. See e.g., the above mentioned Paris, Rizzo PRE 87, 012101 (2013) as well as Leuzzi, Rizzo PRB 109, 174211 (2024). This stems from the structure of the superfield built in Ref. 39/48 by Kurchan.

Response: We have added the important reference PRB 75 144301 (2007) and confirm that it is consistent with our Ansatz. The introduction of Sec. 3 has been rephrased to clarify what we meant when discussing discrepancies between the two approaches. With equilibrium, we always refer to thermal equilibrium. We have made this explicit and now state “Consequently, the aging dynamics of the spherical p-spin model never reaches thermal equilibrium.”

11) Line 321, Eq (23). Please mention that the sigma’s are Pauli matrices.

Response: Done.

12) Section 3.1 Line 334-5. The authors write: “An important distinction between the new platforms and classical glasses is the finite lifetime of the excited states due to spontaneous emission”. How long is such lifetime? Is it long enough for aging to be experimentally detected?

Response: Indeed, one of the main advantages of Rydberg states is their exceptionally long lifetime (reaching into the millisecond range) compared to the inverse of their interaction strength which is typically of the order of a microsecond. Both of these statements are strongly dependent on the specifics of the experimental realization. More highly excited Rydberg states are more stable with stronger interactions, but require more precise experimental control. Although we are not aware of any experiments that have observed aging, there is no fundamental issues that need to be overcome. In fact, Signoles et al. PRX 11, 011011 (2020) have reported anomalously slow decay of the magnetization analogous to the behavior of classical glasses. We have included a statement clarifying the role of decoherence in experimental realizations with Rydberg atoms in Sec 6, where the generalization to open systems is discussed.

13) Eq. (24). Please specify all the elements in the action: m, h_\sigma, etc.. 
The author might consider a Greek index different from \sigma that takes the values +,- to denote the branch of the contour.
Indeed, \sigma denote the Pauli matrices here.

Response: Comments regarding the interpretation of longitudinal field with contour index h_\eta and the effective mass m have been added. We have furthermore replaced the index \sigma with \eta.

14) Eq. (31). What does the superscript V stand for?

Response: It is the Vanishing component, we now explicitly point to this.

15) Line 425 In Eq. (24), in order to describe the critical behavior the authors move from Pauli matrices to soft spins S, i.e., scalar commuting variables in a quartic potential. However, in s_\kappa the Pauli matrices stay and are traced over. Some easy going description might perhaps help the reader to follow the manuscript without compulsorily resorting to ref. 45.

Response: The Pauli matrices \sigma now refer exclusively to Keldysh space, not spin space. To prevent confusion, we have now denoted the Pauli matrices in spin space (“qubits”) by X,Y, Z in Eq (25). We have also added discussion above (26) on how we move from spins to a soft-spin scalar field S.

16) Line 491-499. How is the ferromagnetic phase mentioned here characterized in terms of the correlation functions involved in Eq. (49)? When talking about instability in this dynamic equations, where slow and fast decomposition is assumed, the authors cite the work of de Almeida-Thouless and the Parisi solution, i.e., the instability of the equilibrium solution. The authors should clarify better what they mean by instability in this dynamic case and how it can be analyzed in the dynamic Keldysh theory context, apart from analogies with the statics.

Response: The ferromagnetic phase is characterized by a persistent magnetization indicated by a constant (non-zero) correlation function in the slow regime. We now state this explicitly under Eq. (49). In a quench scenario the condition of thermodynamic stability is replaced by constraints in the form of initial conditions. However, following the Wigner approximation we find all solutions that are accessible by any quench protocol, not all of which have to be thermodynamically stable. Of particular interest are those that are thermodynamically stable, such as the RSB solution. The stability analysis is then identical to that of PRB 75 144301 (2007).

17) Lines 518-520: the authors write “Physically, we can say that replica symmetry breaking corresponds to the inability of the system to fully thermalize to a single global inverse temperature β even at arbitrarily late times/the steady state.” Is the sentence complete? Are the authors trying to say that if a RSB occurs then there is no single state equilibrium phase? Equivalently, that RSB signals multi-equilibria phases?

Response: There was a comma missing after \beta and an in after the slash. Both have been added. Otherwise the sentence is complete. The rephrasing of the referee is also correct.

18) Lines 542-544 In Ref. [75] Derrida introduces the Hamiltonian of the random p-spin model with Ising spins and shows that in the limit p->\infty it is equivalent to the random energy model. This was further discussed by Gross and Mozart, Nucl. Phys. B240, 431 (1988). The soft spin version of the Ising p-spin model introduced by Derrida was introduced and discussed by Kirkpatrick and Thirumalai Ref[21]. The spherical p-spin spin glass model was first introduced in Ref. [77].

Response: We thank the referee for spotting this incorrect reference. It has been fixed. We now correctly attribute the spherical p-spin model to Crisanti and Sommers. We now also more clearly distinguish between the classical and quantum model.

19) Line 545. The SG-PM transition in the spherical p-spin model is not a first order one in the thermodynamic sense: there is no latent heat and no phase coexistence. The order parameter has a discontinuous transition and a dynamic transition (Ref [35]) arises next to the static one. This transition has been termed Random First Order Transition by Wolynes and coworkers in the 80's, including Ref. [21].

Response: Indeed, this statement concerning the quantum model should not have appeared without the corresponding qualifier. This has been fixed.

The authors should comment somewhere about how the Keldysh formalism for spin-glass dynamics is related to the DeDominicis-Peliti-Martin-Siggia-Rose dynamic mean-field theory. For instance in Sec. 4.1.

Response: See the reply to 5).

20) Line 586. “..and in the case of ΣK non-negative.” Sentence unclear, please complete.

Response: The statement has been clarified.

21) Lines 627-628. The authors should clarify what it is meant by saying that the spin-glass phase is indistinguishable from a ferromagnet, in connection to mode coupling theory.
In glass forming liquids at the Mode Coupling temperature the correlation of density fluctuations does not decay to zero anymore but density fluctuations have, nevertheless, still zero mean. Here the analogy should be: non trivial non-zero spin-spin correlation at long times but zero magnetization. Or it is something else occurring?

Response: Our statement only refers to the behavior of correlations and response functions as the average magnetization is inaccessible in the effective action of Q, i.e., after integrating out the original soft spin variables. We now say so explicitly.

22) Lines 638-642 The connection between dynamics and statics has been discussed in Crisanti. Leuzzi PRB 75 144301 (2007) where it is shown that the CHS dynamics of Ref [35] generalized to any number of relaxation (diverging) times scale does lead asymptotically to the static RSB in the Parisi scheme. See also Crisanti de Dominics J. Phys A: Math. Theor. 44 115006 (2011) where it is shown that the static susceptibility in the Parisi RSB scheme coincides with the static limit of the CHS dynamics.

Response: We did not intend to cause the impression that the relation between the result of our dynamical theory and the equilibrium dynamics differs or is superior to results previously reported in the literature. To ensure that there is no misunderstanding, we now explicitly state that the same relation is obtained in by Crisanti and Leuzzi in PRB 75 144301 (2007).

23) Line 721. About experimental realisations. What are the experimental time-scales in the experiments with dephasing? Is it practically feasible to probe the aging regime in those systems?

Response: See the reply to 12).

24) Eventually it is important to notice that reparametrization invariance is proved untrue as soon as the model is not “pure”, that is when more than a single p-spin interaction term is present in the spin-glass Hamiltonian. This is never mentioned throughout the manuscript where only the pure p-spin model is considered but is a crucial issue of critical slowing down and aging in spin-glass systems. See “Rethinking Mean-Field Glassy Dynamics and Its Relation with the Energy Landscape: The Surprising Case of the Spherical Mixed 𝑝-Spin Model" by Folena, Franz, Ricci-Tersenghi PRX 10, 031045 (2020)

Response: The absence of reparametrization invariance is indeed an interesting modification which we have not yet considered. We now mention this at the end of section 5.1 and leave it to future publications to work out how the time-translation-invariant Keldysh formalism needs to be adapted.

25) In discussing the p-spin model the Authors should mention that within the 1RSB solution there is no ultrametricity.

Response: The referee's statement seems too general. As long as the Parisi function is monotonically decreasing, the replica space possesses an ultrametric structure. It is the Keldysh Green’s function $G^K(t)$, which is not ultrametric. This is already stated in the second to last paragraph of Section 4.2.

---

## Round 1 · Referee Report · Anonymous (Referee 2) · 2024-8-15

Strengths

1)General approach and insightful discussion of long-time dynamics

Weaknesses

1) Not clear the novelty and the position of this work in the current literature. 2)Not clear the physical setting and the generality of the results 3)Section 2 very hard to follow without proper definitions.

Report

The manuscript focuses on mean-field spin glasses and discusses how the long-time limit in the Keldysh formalism allows to recover the replica symmetry breaking structure of equilibrium and in particular the ultrametricity.

After an introduction to the problem, the Authors present a general discussion of equilibrium and dynamics and present the conditions under which the two descriptions provides the same physics. Then they apply their framework to two specific mean-field spin glasses, a bosonic version of the Sherrington-Kirkpatrick model and the spherical p-spin model, overall recovering results obtained in equilibrium.

There are several major issues with the current manuscript that the Authors should address.

First, it is not fully clear what is the novelty of the results discussed here. The formal analogy between static and dynamics in mean-field spin glasses is well known in the literature for several years, at least for the classical case.

(the Authors should largely improve their coverage of literature in the Introduction:
two key references to add are:
S. Franz, M. Mézard, G. Parisi and L. Peliti, Measuring equilibrium properties in aging systems, Phys. Rev. Lett. 81, 1758 (1998), doi:10.1103/PhysRevLett.81.1758.
[14] S. Franz, M. Mezard, G. Parisi and L. Peliti, The response of glassy systems to random perturbations: A bridge between equilibrium and off-equilibrium, J. Stat. Phys. 97, 459 (1999), doi:10.1023/A:1004602906332.).

Indeed upon reading the manuscript in detail it seems that many of the key results of this work have been already obtained elsewhere (in particular: Refs 5,6, 48).

In the quantum case perhaps this analogy has not been spelled out in detail (even though in Refs 36, 41, 43 there are many of the key points) but there are in fact no major differences from the classical case. Indeed the use of Keldysh formalism here would be completely equivalent to Martin-Siggia-Ros-Janssen (MSRJ) path integral: the authors never discuss the structure of the quantum vertex and whether these are relevant: their treatment of Keldysh at the level of Dyson equation is completely equivalent to MSRJ. Plus there is no discussion of quantum effects in spin glasses and the effective thermalization ansatz is classical.

Furthermore, the protocol under investigation is not clear at all. The Authors mention quenches: is this unitary dynamics of an isolated quantum glass, or a dissipative dynamics in presence of a bath as usually done in the classical case? What equilibrium solution do they recover from the dynamics? An effective temperature due to a quantum quench or the asymptotic bath temperature?

Finally, the clarity of the manuscript should be largely improved. Section 2 in particular is very hard to follow since several key quantities are not defined (Green's functions, Self-Energy, spin s, ...)

Based on these major points and the requested changes below I cannot recommend this manuscript for publication.

Requested changes

1)The state of the art in the introduction should be largely rewritten to take into account modern literature on the problem. 2)The position and novelty of this work with respect to what is known should be clearly spelled out. 3)The clarity of the manuscript (notation etc) should be largely improved. Section 2 in particular is very hard to follow without proper definitions of the objects which are introduced. 4)The Authors talk about Sherrington-Kirkpatrick, but in the thermodynamic limit this would correspond to a quantum spin in a dynamical field (see for example Ref. 47). Here the authors consider instead a bosonic version with soft-spins. The Authors should comment more extensively on the differences between the two models and the regime of validity of the soft-spin approximation. 5) The SK model (nor the bosonic version under consideration here) cannot be solved exactly even in the mean-field limit, since the resulting quantum impurity model is interacting, differently from the p-spin model. This is why the treatment of Sec.3 is perturbative in the non-linearity. The Authors should clarify this point in the manuscript. 5) The spherical p-spin model, both classical and quantum, has been solved and discussed extensively before Ref. 46. The Authors should properly acknowledge the literature when discussing Sec. 4

Recommendation

Ask for major revision

  • validity: good
  • significance: low
  • originality: low
  • clarity: low
  • formatting: reasonable
  • grammar: reasonable

Author:  Johannes Lang  on 2024-09-04  [id 4737]

(in reply to Report 2 on 2024-08-15)

Referee: The manuscript focuses on mean-field spin glasses and discusses how the long-time limit in the Keldysh formalism allows to recover the replica symmetry breaking structure of equilibrium and in particular the ultrametricity.

After an introduction to the problem, the Authors present a general discussion of equilibrium and dynamics and present the conditions under which the two descriptions provides the same physics. Then they apply their framework to two specific mean-field spin glasses, a bosonic version of the Sherrington-Kirkpatrick model and the spherical p-spin model, overall recovering results obtained in equilibrium.

There are several major issues with the current manuscript that the Authors should address.

First, it is not fully clear what is the novelty of the results discussed here. The formal analogy between static and dynamics in mean-field spin glasses is well known in the literature for several years, at least for the classical case.

(the Authors should largely improve their coverage of literature in the Introduction: two key references to add are: S. Franz, M. Mézard, G. Parisi and L. Peliti, Measuring equilibrium properties in aging systems, Phys. Rev. Lett. 81, 1758 (1998), doi:10.1103/PhysRevLett.81.1758. [14] S. Franz, M. Mezard, G. Parisi and L. Peliti, The response of glassy systems to random perturbations: A bridge between equilibrium and off-equilibrium, J. Stat. Phys. 97, 459 (1999), doi:10.1023/A:1004602906332.).

Indeed upon reading the manuscript in detail it seems that many of the key results of this work have been already obtained elsewhere (in particular: Refs 5,6, 48).

In the quantum case perhaps this analogy has not been spelled out in detail (even though in Refs 36, 41, 43 there are many of the key points) but there are in fact no major differences from the classical case. Indeed the use of Keldysh formalism here would be completely equivalent to Martin-Siggia-Ros-Janssen (MSRJ) path integral: the authors never discuss the structure of the quantum vertex and whether these are relevant: their treatment of Keldysh at the level of Dyson equation is completely equivalent to MSRJ. Plus there is no discussion of quantum effects in spin glasses and the effective thermalization ansatz is classical.

Furthermore, the protocol under investigation is not clear at all. The Authors mention quenches: is this unitary dynamics of an isolated quantum glass, or a dissipative dynamics in presence of a bath as usually done in the classical case? What equilibrium solution do they recover from the dynamics? An effective temperature due to a quantum quench or the asymptotic bath temperature?

Finally, the clarity of the manuscript should be largely improved. Section 2 in particular is very hard to follow since several key quantities are not defined (Green's functions, Self-Energy, spin s, ...)

Based on these major points and the requested changes below I cannot recommend this manuscript for publication.

Response: We thank the referee for the recommended modifications, which we have considered in detail. Please find below our responses to the individual issues raised.

Indeed, our paper provides a lot of historical background as we aim for it to be accessible to a wide audience. As such, it appears that to expert readers, the distinction between old and new results has been eroded in our previous version. We now make a much clearer statement regarding the novelty of our results. These are, in particular, the introduction of a new and explicit connection between the Dyson-Keldysh equations of quantum systems and the algebra of Parisi matrices. In addition, as correctly pointed out by the referee, our paper provides the first complete discussion of the dynamics of the mean-field quantum Ising spin glass with full replica symmetry breaking.

Concerning the specific references mentioned:

Both Ref. 5 and 6 discuss the late-time dynamics of the classical Sherrington-Kirkpatrick model and spherical p-spin model while keeping track of the effects of the initial quench at all times. As opposed to this, we are interested in the universal results independent of the specifics of the quench protocol and thus make simplifications that restore time-translation invariance.

Ref. 48 indeed achieves a similar relation between the supersymmetric and replica formalisms. However, the introduction of supersymmetry requires significant analytical overhead, that can be avoided in the analogy of the replica approach with the dynamical Keldysh formalism. Since both of these methods are largely independent with different strengths and weaknesses, we consider both Ref. 48 and our paper important contributions to the field.

Since we are dealing with quantum systems, we consider quenches of isolated systems, although the addition of a bath does not change the qualitative results but merely fixes the inverse temperature \beta. In the absence of a bath, the effective temperature is set by the quench protocol. We agree that this was not sufficiently clear in the text and now state so explicitly in Sec. 2.2.

Although the solution to the quantum model turns out to be a classical spin glass at any finite temperature or quench strength, it is by no means clear that this is the only allowed solution. Our analysis thus provides an important starting point for the investigation of quantum spin glasses with the same scaling behavior for G^K and G^R as pointed out in Sec. 5.2. We thus maintain that although the same results could have been obtained by an MSRJD path integral, their validity/possible violations can only be checked by the Keldysh formalism discussed here.

We have now rewritten the abstract and made significant changes to Sections 1 and 2 to make the novelty of our work and its connections to the earlier work clearer. Taking into account the comments of both referees, we have put great emphasis on an unbiased and more complete account of the vast literature. In doing so, we have also added the two references mentioned above. Furthermore, we are now much more pedagogical in the introduction of the dynamical formalism in Sec. 2.

Referee: 1)The state of the art in the introduction should be largely rewritten to take into account modern literature on the problem.

Response: As stated above, the introduction has been rewritten to provide a more complete account of the literature, taking into account the comments of both referees.

Referee: 2)The position and novelty of this work with respect to what is known should be clearly spelled out.

Response: This has been taken care of in the rewriting of Secs. 1 and 2. See also our previous comments.

Referee: 3)The clarity of the manuscript (notation etc) should be largely improved. Section 2 in particular is very hard to follow without proper definitions of the objects which are introduced.

Response: Sec. 2.2 now provides a much more pedagogical introduction to Keldysh formalism with a great emphasis on clarity.

Referee: 4)The Authors talk about Sherrington-Kirkpatrick, but in the thermodynamic limit this would correspond to a quantum spin in a dynamical field (see for example Ref. 47). Here the authors consider instead a bosonic version with soft-spins. The Authors should comment more extensively on the differences between the two models and the regime of validity of the soft-spin approximation.

Response: We presume the referee is referring to the first-order time derivative Berry phase terms that accompany quantum spins but not quantum rotors. There is an unfortunate confusion in the spin glass literature, where quantum rotor models have been labeled as quantum spin models. However, for Ising spins in the presence of a transverse field, quantum spins do indeed become quantum rotors: in this case, we can integrate the transverse components out, and express the effective action in terms of the Ising order parameter only. This effective action has only second-order time derivatives, as is the case for quantum rotors. This has now been noted above Eq (26).

Referee: 5) The SK model (nor the bosonic version under consideration here) cannot be solved exactly even in the mean-field limit, since the resulting quantum impurity model is interacting, differently from the p-spin model. This is why the treatment of Sec.3 is perturbative in the non-linearity. The Authors should clarify this point in the manuscript.

Response: Indeed, the discussion of the SK model is perturbative and thus should not be expected to provide quantitatively accurate results, especially at short times where correlations are short-ranged. However, the perturbative discussion produces all RG-relevant terms necessary for a qualitative description of the spin-glass transition. This statement was first made by Read et al. in Phys. Rev. B 52 384 and is now echoed in the revised Sec. 3.3.

Referee: 6) The spherical p-spin model, both classical and quantum, has been solved and discussed extensively before Ref. 46. The Authors should properly acknowledge the literature when discussing Sec. 4

Response: The introduction of Sec. 4 has been extended. We apologize for the previous omissions of important references in this section.

---

## Round 2 · Referee Report · Anonymous (Referee 1) · 2024-10-21

Report

The Authors replied positively to the comments of previous report. It can be published in the present form.

Recommendation

Publish (easily meets expectations and criteria for this Journal; among top 50%)

---

## Round 2 · Referee Report · Anonymous (Referee 2) · 2024-10-25

Report

Dear Editor,
I have read the new version of this manuscript as well as the Author response. I am glad to see that the Authors have followed the comments by the two Referees and largely revised their manuscript. It is now more clear how to place this work in the literature on classical and quantum spin glasses and what are the major novelties. Although one could argue that this work lacks major groundbreaking new results and that to a certain extent most of its content was known to the community (at least on the classical side), I still believe there is a benefit in presenting the material as it is done in this work for quantum spin glasses. Overall I think the manuscript can be accepted for publication, even though SciPost Physics more restrictive acceptance criteria, as opposed to SciPost Core, do not seem to be fully met here.

Recommendation

Publish (meets expectations and criteria for this Journal)

---

## Round 2 · List of Changes

Clarified novelty of results and relation to previous work in abstract and Sec. 1
Significantly extended the list of references to more accurately represent the current state of the field
Extended Sec. 2.2 to increase the accessibility of Keldysh formalism used in the manuscript and clarify the physical protocol.
Numerous changes to notation and phrasing to enhance readability

---

## Editorial Decision

published